# Glaciological settings and recent mass balance of Blåskimen Island in Dronning Maud Land, Antarctica

Vikram Goel[1], Joel Brown[1,2], and Kenichi Matsuoka[1]

[1]Norwegian Polar Institute, Tromsø, Norway
[2]Aesir Consulting LLC, Missoula, Montana, USA

*Correspondence to:* Vikram Goel (vikram.goel@outlook.com)

**Abstract.** The Dronning Maud Land coast in East Antarctica has numerous ice rises that very likely control the dynamics and mass balance of this region. However, only a few of these ice rises have been investigated in detail. Here, we report field measurements of Blåskimen Island, an isle-type ice rise adjacent to the Fimbul Ice Shelf. Blåskimen Island is largely dome shaped, with a pronounced ridge extending to the southwest from its summit (410 m a.s.l.). Its bed is mostly flat and about 100 m below the current sea level. Shallow radar-detected isochrones dated with a firn core reveal that the surface mass balance is higher on the southeastern slope than the northwestern slope by ∼37%, and this pattern has persisted for at least the past decade. Arches in radar stratigraphy suggest that the summit of the ice rise has been stable for ∼600 years. Ensemble estimates of the mass balance using the input-output method show that this ice rise has thickened by 0.12–0.37 m ice equivalent per year over the past decade.

## 1 Introduction

Around 74% of the Antarctic coastline consists of floating ice shelves fed by outlet glaciers and ice streams (Bindschadler et al., 2011). These ice shelves together with numerous pinning points (ice rises and rumples) regulate the outflow of the grounded ice (Dupont and Alley, 2005; Matsuoka et al., 2015; Fürst et al., 2016). Embedded into the ice shelf, ice rises create a zone of compression upstream of the ice rise which buttresses the ice shelf (Borstad et al., 2013). However, downstream of the ice rise the tensile forces leave a weak region subject to crevasses and thinner ice shelves (Favier and Pattyn, 2015). Ice rises strongly influence regional surface mass balance (SMB) (Lenaerts et al., 2014) and can significantly alter the timing of deglaciation of the ice sheet (Favier and Pattyn, 2015; Favier et al., 2016). Although relatively small in footprint, ice rises can have far-reaching effects on the Antarctic ice sheet dynamics.

Ice rises are also a useful resource for investigating evolution and past climate in the coastal region. Englacial (isochronous) stratigraphy detected using radar has been widely used to constrain the evolution of the ice rise and adjacent ice body over the past millennia (Conway et al., 1999; Nereson and Waddington, 2002; Hindmarsh et al., 2011; Siegert et al., 2013; Drews et al., 2015; Kingslake et al., 2016). The knowledge of the evolution of an ice rise is crucial to retrieve reliable past regional climatic changes, using ice cores drilled through it (Mulvaney et al., 2002, 2014).

The 2000 km long coastline of Dronning Maud land (DML, 20°W–45°E), East Antarctica, consists of numerous outlet glaciers and ice shelves punctuated by some 30 ice rises (Matsuoka et al., 2015). These ice rises most likely contribute significantly to the shape and mass balance of this region. In addition, they present an opportunity to better understand the evolution of this data sparse region (Mackintosh et al., 2014). So far, only two ice rises have been investigated in DML: Derwael Ice Rise (26°E) in Roi Baudouin Ice Shelf (Drews et al., 2015) and Halvfarryggen Ice Dome (6°W) between Jelbartisen and Ekstromisen ice shelves (Drews et al., 2013). Both ice rises are grounded on flat bed ~200 m below the current sea level, show large SMB contrast across upwind/downwind slopes and have dynamic characteristic times of hundreds of years. Stratigraphic evidence shows that these ice rises have been in nearly steady state over the last several millennia (3000–5000 years). These ice rises are separated by ~1200 km along the coast, where glaciological settings are variable. For example, along the DML coast between the two ice rises, SMB can vary by at least a factor of two, depending on surface topography, storm tracks and wind direction (King, 2004; Lenaerts et al., 2014). This region also consists of several ice shelves and outlet glaciers with flow speeds varying by a factor of four (Rignot et al., 2011). Ice rises in these varying settings can evolve and impact adjacent ice shelves differently and hence needs to be investigated.

We carried out field measurements of Blåskimen Island, an isle-type ice rise located west of the Fimbul Ice Shelf at the calving front (Fig. 1a), to elucidate the current status and past evolution of this coastal region. Here, we present surface and bed topography, surface flow field, and surface mass balance of Blåskimen Island. Our analysis of these data implies that the ice rise has thickened by 0.12–0.37 m a$^{-1}$ (ice equivalent) over the past decade.

## 2   Blåskimen Island

Blåskimen Island has a total area of 651 km$^2$ (Fig. 1a; Moholdt and Matsuoka, 2015). It is located between the Fimbul and Jelbart Ice Shelves near the calving front. Both of these ice shelves flow around several isle-type ice rises (isolated from the ice sheet by an ice shelf) and promontory-type ice rises (elongated extension of the ice sheet into an ice shelf). The Fimbul Ice Shelf is mainly fed by the Jutulstraumen Glacier, one of the largest outlet glaciers in DML (Høydal, 1996). The Jelbart Ice Shelf is fed by the Schytt Glacier, which is slower by a factor of two and narrower, than the Jutulstraumen Glacier (Rignot et al., 2011). Blåskimen Island, along with two more ice rises upstream, Novyy Island and Lejtenanta Smidta, separate these two ice shelves. The ice between Blåskimen Island and these ice rises moves very slowly ($\ll$ 10 m a$^{-1}$ Rignot et al., 2011). In addition, there are four smaller ice rises and rumples near the calving front of the western shear margin of the fast-flowing portion of the Fimbul Ice Shelf (Moholdt and Matsuoka, 2015). Thus towards east, Blåskimen Island is surrounded by much slowly moving ice whereas it abuts the eastern shear margin of the fast flowing part of the Jelbart Ice Shelf towards west.

## 3 Field measurements and data processing

We carried out field surveys of Blåskimen Island during the austral summers of 2012–2013 and 2013–2014. It included kinematic and static GPS surveys (Section 3.1), shallow- and deep-sounding radar profiling (Section 3.2), as well as firn coring and borehole temperature measurements (Section 3.3). The location of these measurements is shown in Fig. 1b.

### 3.1 Kinematic and static GPS surveys

To develop digital elevation models (DEM) of the ice rise surface, we conducted kinematic GPS surveys using Trimble dual-frequency receivers. Two units were installed near the ice-rise summit, one acting as a base station, with one for redundancy. Five rover stations were mounted on snowmobiles, which moved at a speed of $\sim$15 km h$^{-1}$. Our survey resulted in surface elevation measurements at the average interval of $\sim$4 m along the survey transects spaced 0.8–1 km from each other. Additional surveys were carried out in the vicinity of the summit to precisely determine the summit position of the ice rise and in the eastern part of the ice rise where satellite imagery shows surface lineations (light grey feature over dark grey in Fig. 1b).

To measure the surface-flow field, we installed 3 m long hollow aluminium stakes at 97 locations on Blåskimen Island. The stakes were installed $\sim$1 m into the snowpack without any anchor resisting vertical motion. Out of these, 55 stakes were installed along 6 steepest descent paths determined with the surface DEM. The other 41 stakes were installed in the vicinity (< 2.5 km) of the summit to better resolve small ice-flow speeds there. The stakes were occupied for $\sim$20 minutes to determine their lateral positions (e.g., Conway and Rasmussen, 2009; Matsuoka et al., 2012b). The position of each of the stakes was measured in January 2013. In January 2014 we remeasured the positions of the 90 of these stakes. Six stakes were buried and one found heavily tilted (more than $\sim$20°). We did not use GPS-measured vertical positions to determine ice-thickness changes, because of possible motion of the stakes relative to the firn and firn densification. Nevertheless, we measured their heights to the snow surface to estimate surface mass balance (SMB) over the year.

Instantaneous kinematic, and average static-rover station locations were determined relative to the base stations for each field season using TRACK software package, part of the GAMIT/GLOBK GPS positioning software (Herring et al., 2010). Base-station positions for each field season were determined using Canadian geodetic Precise Point Positioning system (CSRS-PPP; https://webapp.geod.nrcan.gc.ca/geod/tools-outils/ppp.php). These base stations moved negligibly over the 5 days of GPS campaigns each year; less than the error of each GPS location ($\sim$1 cm laterally). To convert heights above the WGS84 ellipsoid to heights above local sea level, we subtracted 13 m of geoid height uniformly provided by GOCE gravity product (https://earth.esa.int/web/guest/data-access/browse-data-products/-/article/goce-gravity-fields-5777).

### 3.2 Ice-penetrating radar profiling

We collected common-offset radar transects along four of the steepest-decent paths concurrent with the GPS stake locations. These radar transects were collected with a 2-MHz ground-based radar system with resistively-loaded dipole antennas (Matsuoka et al., 2012a) to reveal the ice thickness and englacial isochronous stratigraphy. We also operated GSSI/SIR3000 radar with 400-MHz antenna to detect stratigraphy within the top $\sim$50 m of the ice rise (Hawley et al., 2014). These two radar surveys

were collected with snowmobiles moving at 8–10 km h$^{-1}$ towing the antennas. The antenna positions were determined using kinematic GPS attached to the snowmobiles. This resulted in an average radar trace spacing of ~5 m for the deep-sounding radar and ~0.25 m for the shallow-sounding radar.

Post processing included using a dewow filter, an Ormsby band-pass filter, and depth-variable gain functions. To calculate ice thickness, we assumed a radio-wave propagation speed of 169 m $\mu$s$^{-1}$ and added a firn correction to account for faster propagation in the firn. We used an optimization inversion routine (Brown and Matsuoka, in prep) to model depth profiles of density along shallow radar profiles using (1) surface densities measured at 13 locations, and (2) depth profile of the density measured along the 23 m long firn core. We then used the depth profiles of the density along the radar profile to estimate laterally variable firn corrections. The magnitudes of the firn correction vary between 4–6 m. To calculate depths of englacial reflectors, we used variable propagation speeds discussed later in Section 4.1.

### 3.3 Firn cores and borehole-temperature measurements

We drilled a 23 m long firn core near the summit of Blåskimen Island and transported the frozen samples to laboratories to develop visual stratigraphy as well as chemical, isotope, and dielectric stratigraphy. The core was dated back to 1996 by counting annual cycles of oxygen isotopes, and by identifying volcanic horizons using non-sea-salt sulphate data (Vega et al., 2016). They found that SMB in the past 17 years ranges between 0.44 m a$^{-1}$ in 2004 and 1.32 m a$^{-1}$ in 2011, giving the mean SMB in this period 0.76 m a$^{-1}$; in this paper, mass balance and SMB are always shown as meters of ice equivalent.

We installed a thermistor string in the borehole and measured temperature profiles from surface to 20 m depth. Within 25 hours after the completion of drilling, the temperature became steady with ± 0.1 °C of variation at each depth. At depths between 8 and 12 m the temperature was ~ -16.2 °C.

Separately, we drilled nine 3 m long firn cores (locations are shown in Fig. 1b) to measure spatial variations of surface density. Hereafter we call mean density in the top 3 m as surface density. It was determined by measuring core volume and weight. Measured surface density through firn cores varies by ± ~2.5% over the ice rise, with a mean value of 453 kg m$^{-3}$ (uncertainty: 3% or 14 kg m$^{-3}$). However, no distinct pattern in surface density variation was observed in terms of elevation or slope direction. When estimating SMB below, we bilinearly interpolated the surface density.

## 4  Analytical methods

### 4.1  Surface mass balance

We estimated surface mass balance (SMB) using two different methods. The first method uses the heights of GPS stakes above the snow surface. The second method for determining SMB uses isochronous radar reflectors profiled with the shallow-sounding radar. Both methods require surface-density distribution measurements.

We estimated SMB at each stake by multiplying the measured stake-height differences by the measured surface density. Considering measurement errors and uncertainty accounting snow densification (Eisen et al., 2008), this estimate has an overall

uncertainty of ± 6%. We consider the sinking of the stake under its own weight to be minimal, as the observed surface densities are high.

Provided that radar reflectors are isochronous (Richardson et al., 1997), SMB can be derived from dated radar reflectors. In this analysis, we assume that the effects of vertical strain (thinning after the deposition of snow) on reflector depths are

negligible so that thickness of an ice layer bounded by the radar reflectors are solely controlled by the differences in SMB (Waddington et al., 2007). Thus shallow-ice approximation can hold, when depth $h$ (ice equivalent) of a radar reflector is much smaller than the local ice thickness $H$ ($h \ll H$). For our case, $h/H$ is less than 0.04 and thus the shallow-ice approximation is in most cases valid.

However, the shallow-ice approximation may not be valid in areas where vertical strain rates are large, such as the region

near an ice-flow divide (Gillet-Chaulet et al., 2011; Kingslake et al., 2014). In this region, accumulated effects of variable vertical strain can result in upward arches in isochrones, so-called Raymond arches (Raymond, 1983), which were found at many other ice rises (e.g. Vaughan et al., 1999; Conway et al., 1999; Nereson and Raymond, 2001). Such upward arches can also be caused by anomalously low SMB near the summit possibly due to wind erosion (Drews et al., 2013, 2015).

Vaughan et al. (1999) demonstrated that amplitudes of upward arches induced by anomalous SMB increases linearly with ice-

equivalent depth, whereas the Raymond effect makes the amplitude increase quadratically. We used this criterion to diagnose the origin of shallow upward arches near the current summit (Fig. 2b). We first derived ice-equivalent depths of reflectors, assuming that firn density does not vary laterally. For this purpose, we used the depth profile of density $\rho(z)$ at the core site. We estimated local propagation speed $v(z)$ at a depth $z$ using a relationship between density and refraction index $n(z)$ (Kovacs et al., 1995):

$$v(z) = \frac{c}{n(z)} = \frac{c}{1 + 0.851\rho(z)} \tag{1}$$

Here, $c$ is the propagation speed of light in the vacuum (300 m $\mu s^{-1}$). Then, we estimated two-way propagation time $t(z)$ to each depth $z$:

$$t(z) = \int\limits_{0}^{z} \frac{2}{v(z)} dz \tag{2}$$

Second, using these ice-equivalent depths $z$, we measured arch amplitudes from the arch top to the baseline defined with

reflector depths 1 km away to both sides of the arch.

Under the shallow-ice approximation, we accounted for density variations using two methods. The first method is to account only for vertical variations in density and ignore its lateral variations. For this purpose, we used the measured densities of the 23 m long core. The second method is to account for spatial variations in density, temperature, and SMB altogether using an inversion method (Brown and Matsuoka, in prep). The former method is not strictly valid, because the surface density varies

by ± ∼2.5% and possible variations in SMB add more complexities in density at depths away from the core site. The latter method is also not strictly valid as it solves for the best fit at all locations to a steady-state firn densification model. Firn-core analysis shows no significant temporal trend but large year-to-year variations by a factor of three over the past 17 years (Vega

et al., 2016). Nevertheless, the model fits the measured density well (within 95% confidence bounds of the fit). Although these two methods use distinct assumptions, we show that they give consistent results and thus more confidence in SMB estimates using inaccurate data representations in these two methods.

## 4.2 Mass balance of the ice rise

We applied the Input–Output (I-O) Method (e.g., Rignot and Kanagaratnam, 2006; Conway and Rasmussen, 2009; Zwally and Giovinetto, 2011) to individual columns over the ice rise (Section 4.2.1). The I-O Method calculates the mass balance as a difference between incoming $Q_{in}$ and outgoing $Q_{out}$ fluxes from all the sides of a column, with SMB ($M_{SMB}$) over the column area added. Here, we ignore basal melting, as one-dimensional thermomechanical model (Neumann et al., 2008) shows no basal melt for the geothermal flux estimated in this region (57 mW m$^{-2}$; Maule et al., 2005). Radargrams show no

anomalous features in the radar reflection from the bed that could indicate basal melting (Fig. 2c).

The ice-flow fluxes through an ice column are calculated as:

$$Q = \rho \gamma u_\perp h \qquad (3)$$

Here, $\rho$ is the density of a column, $\gamma$ is a dimensionless factor which scales the measured surface flow speed $u_\perp$ normal to the gate to depth averaged speed $u_{av\perp}$; $u_{av\perp} = \gamma\, u_\perp$, with $\gamma \leq 1$. This implicitly assumes that ice-flow direction does not change with depth, which seems valid over relatively flat bed terrain underneath the ice rise (Section 5.1).

### 4.2.1 Constraining $\gamma$

The parameter $\gamma$ is a function of local surface slope, ice thickness, ice temperature and ice rheology (Cuffey and Paterson, 2010). For isothermal ice flow over non-sliding bed and assuming shallow-ice approximation $\gamma$ can be stated as $\gamma = \frac{(n+1)}{(n+2)}$, where $n$ is the creep exponent of Glen's flow law (Cuffey and Paterson, 2010, p.310). Previous studies on ice-flow divides

suggest that $n$ can be between 3 and 5 (Martín et al., 2009a, b; Gillet-Chaulet et al., 2011; Drews et al., 2015). Then $\gamma$ ranges between 0.8 ($n = 3$), and 0.86 ($n = 5$).

The actual range of $\gamma$ can be different, because the isothermal approximation is hardly valid. Also, near the ice-flow divide the shallow-ice approximation is invalid as longitudinal stresses are significant (Raymond, 1983).

Reeh (1988) showed that, near the flow divide, $\gamma$ can be close to 0.5, when $n = 3$ and ice is isothermal. As ice becomes warmer

at greater depths, the deeper ice is presumably softer than the shallower ice, implying that $\gamma$ is larger when ice-temperature variations are considered. Hence, it is reasonable to assume that $0.5 \leq \gamma \leq 1$. Raymond (1983) used an isothermal model, and Hvidberg (1996) used a thermomechanical model to constrain the range of $\gamma$ near the divide. Both showed that $\gamma$ is smallest at the divide, and varies largely near the divide region.

Out of the divide region several ice thicknesses from the divide, $\gamma$ becomes less variable. For isothermal two-dimensional

(divide) flow, $0.61 \leq \gamma \leq 0.75$ (< $8H$ from the divide, Raymond (1983)) and $0.56 \leq \gamma \leq 0.77$ (< $10H$ from the divide, Hvidberg (1996)). For themo-mechanical case, $0.69 \leq \gamma \leq 0.86$ (< $10H$ from the divide, Hvidberg (1996)) and for isothermal axisymmetric radial flow $0.54 \leq \gamma \leq 0.76$ (< $10H$ from the divide, Hvidberg (1996)). Although radial flow leads to an increase

in the divide region, $\sim 70\%$ of the changes in $\gamma$ still happen within 4 ice thiknesses from the divide. Therefore, for the setups discussed below (with an average extent of $\sim$9 km or 18 ice thicknesses from the divide), $\gamma$ remains virtually uniform.

In the study, we consider the spatially uniform $\gamma$ in each estimate and examine ensemble results for within the plausible range (0.7–0.9). More accurate determination of $\gamma$ requires the knowledge of ice-rise evolution in the past millennia, because
ice rheology has a memory through ice temperature and crystal fabric. This requires detailed ice-flow modelling, which is beyond the scope of this study.

### 4.2.2  Estimate setups

We estimated mass balance for three different setups: (i) polygons bounded by GPS stakes, (ii) uniformly-distributed square columns (grid) on the ice rise and average them over individual polygons for (i), and (iii) several flowbands along GPS stakes
and radar profiles. The polygon and grid setups have good spatial coverages, but they require data interpolations to large degrees. The flowband setup relies more on direct field measurements, but has very limited spatial coverage. Therefore, we use these three mass balance estimates as an ensemble.

#### 4.2.2.1  Polygon setup

As ice rises are expected to show slope dependent SMB features (King, 2004; Lenaerts et al., 2014), it is probable that mass
balance could also have similar features. To account for this, in this method we divide the ice rise into 19 polygons in respect to the surface slope direction and data availability. All four sides of these polygons act as a flux gate. Ice thickness and surface-flow velocities are available at each corner of these polygons. We observe some cases that ice is thicker at one corner but ice flows faster at the other corner. Therefore, to better account for these variations along the gate, we divided each gate into ten subgates and estimated the flux at each subgate, rather than calculating mass flux using single values averaged over an entire
gate.

#### 4.2.2.2  Grid setup

We divide the ice rise into grid with 200 m long square columns and estimated mass balance for each column. The mass balance values are then averaged over each polygon used in the polygon setup. We adopt this setup, together with the polygon setup, to test how data interpolations affect the mass balance estimate. We calculated mass balance $MB$ using the continuity equation
to each grid element:

$$MB = \frac{\partial h}{\partial t} = M_{SMB} + (\frac{\partial u}{\partial x} + \frac{\partial v}{\partial y})H + (u\frac{\partial h}{\partial x} + v\frac{\partial h}{\partial y}) \tag{4}$$

Where $u$ and $v$ are the components of ice-flow vectors in the rectangular (local coordinate) directions $x$ and $y$, and $h$ is the variable for ice thickness. We bilinearly interpolate measured $u$, $v$, $M_{SMB}$, and $H$ into grids.

#### 4.2.2.3 Flowband setup

In this method, we calculate mass balance along ice-flow bands of varying widths in three different slopes of the ice rise. We define a flowband width to account for flow divergence and convergence along the flowband, assuming that ice flows along the steepest descent path on the surface. Flowband width at the downslope end is taken as 1 km, and for each flowband, steepest ascent paths are determined from two points 0.5 km away from the most downstream GPS stake. We used ascent because the surface topography near the summit is much less distinct and consequently the divergence estimate is more sensitive to small topographic changes. We rejected three flow profiles out of six, because the GPS markers were not within the defined flowband. Along the three flowbands, the flowband widths vary by a factor of 1.4–3.6. This variation depends on the initial band width (1 km used here), but over the range of the initial band width between 0.9 and 2.5 km, the band width estimates vary only ∼3%. We further divided the flowbands into three columns based on the available data and calculated mass balance similarly to the polygon setup.

## 5 Results

### 5.1 Topography and surface-flow field

Figure 3a shows surface elevations derived from kinematic GPS surveys using bilinear interpolation. The summit is 410 m a.s.l. and is ∼350 m higher than the surface of the ice shelf on the southern side (ice-shelf elevation is taken from Fretwell et al. (2013)). From the summit, the elevation drops gradually towards the edges of our survey region in all directions, giving a relatively dome-shaped topography to the ice rise. A pronounced ridge extends from the summit to the southwest. The eastern flank shows locally steep slopes and a basin in northeast with overall lower, less-tilted surface. The eastern steep slopes and the southwest ridge are consistent with lineations observed in satellite imagery (light gray feature over dark gray in Fig. 1b). Along a profile through the summit (2–2' in Fig. 1b), the absolute surface slope smoothed over 500 m long segments ranges between 0.02 and 0.04, except for the summit vicinity of ∼1 km where the surface is virtually flat (Fig. 2a).

The radar profiles visualized the bed as well as ice stratigraphy (Figs. 2b and 2c). The measured ice thickness of Blåskimen Island ranges between 374 m (first quartile) and 444 m (third quartile), with a mean value of 400 m. The ice becomes thinner gradually in all directions away from the summit. Considering uncertainty associated with digitization of the bed reflector, data sampling and firn correction, the uncertainty in the ice thickness is ± 5 m.

We derived the bed elevations by subtracting the ice thicknesses from the surface elevations. At locations where radar data are available, the bed elevation is on average 110 m below the current sea level (-110 m a.s.l.), ranging between -68 m a.s.l. and -125 m a.s.l. (first and third quartiles, respectively). The highest point (-22 m a.s.l.) on the bed was observed about 6 km northeast of the summit (along the 4–4' profile, see Fig. 1b). We developed a bed DEM using bilinear interpolation (Fig. 3b). The bed of the central part of ice rise is very flat; in this region bed elevations vary only by ∼50 m within an area of ∼100 km$^2$. Also, individual radar profiles show that this region is smooth (Fig. 2c). This low, flat, and smooth region extends from the summit vicinity towards north and northwest, and constitutes the majority of the ice-rise bed. However, towards the

southern end of the survey domain, the bed elevation decreases by ~200 m over a horizontal range of ~5 km, resulting in a mean bed slope of ~0.04. Another steep bed (0.03–0.04) is found in the northeastern slope. These steeper regions in the bed are associated with steeper regions on the surface, although the surface is not as steep as the bed.

The surface-flow field is shown in Fig. 3c. The GPS stakes within 2 km of the summit moved only negligibly ($< 0.1$ m a$^{-1}$). The displacement of stakes outside the summit region is larger and increases downstream; ice flows less than 3 m a$^{-1}$ within 4 km from the summit and 10–15 m a$^{-1}$ downslope. Ice flows slowest along the ridge towards southwest, whereas the fastest flow is along the south section of flowline 2–2'. The estimated positions have a mean precision of 4.9 cm and 5.1 cm for the east/west and north/south components leading to a processing uncertainty on the velocity of $\pm 7$ cm a$^{-1}$. This does not include uncertainty associated with any tilt of the stakes. Nevertheless, as the velocities outside the summit area range between 4 m a$^{-1}$ and 15 m a$^{-1}$, and the observed tilts were small, we consider this uncertainty to be negligible.

## 5.2 Surface mass balance

The mean SMB from 90 stake-height measurements across Blåskimen Island is 0.78 m a$^{-1}$, for the period between January 2013 and January 2014 (Fig. 4a). SMB varies by a factor of 3.3 over the study area of ~20 km by ~20 km (0.28–1.03 m a$^{-1}$), with 80% of values ranging between 0.69 and 1.03 m a$^{-1}$. The SMB shows a distinct spatial pattern; it is larger in the southern slope and smaller in the northern slope. The surface is rougher (sastrugi) in the low SMB region whereas smoother and softer in the high SMB region, indicating strong impact of the wind. The summit vicinity has a large number of stakes, which show small variations of SMB without any distinct pattern.

The shallow-sounding radar visualizes continuous reflectors within the firn (Fig. 2b). No major disruptions are observed that can be associated with surface melt or strong wind scour. Therefore, we assume these continuous reflectors are isochrones (Richardson et al., 1997). With this assumption, we can associate firn-core ages to radar reflectors. We tracked three reflectors to almost the full extent of the radar surveys; at the 23 m long core site, they are at 8.4 m (actual depth or 2.15 m ice-equivalent depth; dated 2011 or 3 years before the survey by Vega et al. (2016)), 11.9 m (4.2 m; 2009), and 12.8 m (6.9 m; 2005).

To judge whether the reflector depths are controlled by SMB or Raymond effect (Section 4.1), we measured amplitudes of arches near the current ice-flow divide. This analysis was made for two deeper reflectors of three that we used for SMB estimates, as the arch amplitude for the shallowest reflector is insignificant. In addition, we measured arch amplitudes of six more reflectors at greater depths. These reflectors range between ~4 m and ~35 m ice-equivalent depth (Fig. 2b). We analyzed the arch amplitudes in this depth range to better resolve their depth variations. All four radar profiles across the summit (Fig. 1b) show that the arch amplitude increases linearly with depth (Fig. 5). Therefore, we conclude that the shallow-ice approximation can be used all along the radar profiles and thus the three radar reflectors within the top ~7 m ice-equivalent represents spatial patterns of SMB.

The 23 m long core shows that the density varies ~36% (450–655 kg m$^{-3}$) vertically along its length and the 13 shallow cores show that the surface density varies $\pm$ ~2.5% horizontally. Amongst three reflectors we analysed, the deepest reflector (12.8 m actual depth at the core site) has the largest depth range between ~8 and ~15 m. It implies that the use of a uniform density could make the SMB estimates less accurate, and variable density should be accounted for.

Figure 2a shows the SMB averaged over the past 9 years (2005–2014) estimated using the two radar methods along the profile 2–2'. The differences between the SMB estimated with these two methods are localized to the region in 1–6 km north (profile 2–2', Fig. 2a) and northeast (along profile 4–4') of the summit. Except for this region, these two methods give nearly identical SMB spatial patterns along the radar profiles. Figure 4b shows the SMB estimated using the first radar method (accounting for the vertical density variations only). It gives the spatially mean value of 0.81 m a$^{-1}$, with the first and third quartiles of 0.71 and 0.93 m a$^{-1}$ respectively. The second radar method accounting for both lateral and vertical density variations gives a slightly lower number than the second radar method by 5–10%; the mean value for the second radar method is 0.75 m a$^{-1}$, with first and third quartiles of 0.65 and 0.85 m a$^{-1}$ respectively.

Factors determining the uncertainty in SMB derived from dated radar isochrones can be broadly categorized as: (1) error in determining the depth of the reflector, (2) error in dating the firn core, (3) error in estimating the cumulative mass above the reflector and its spatial variability. For the first source we assess the uncertainty to be within ± 10 cm. We consider the combined errors in depth-age scale and error in linking it to radar reflector to be ± 1 year. The density model used to fit the observations has an uncertainty of ± 3%, whereas we see ± 3% variability in the surface density. Using standard error propagation, this results in an uncertainty of ± 11%. It is larger than an uncertainty of ± 6% for the stake method.

## 5.3 Mass balance

We estimated mass balance over the nine-year period between 2005 and 2014 (Fig. 6). The flowband setup shows a mean mass balance of +0.12 ± 0.10 and +0.27 ± 0.10 m a$^{-1}$ over the range of $\gamma$. The uncertainty (± 0.10 m a$^{-1}$) is estimated using the uncertainties in ice thickness (± 5 m), flow speed (± 7 cm a$^{-1}$), ice density after correcting for firn (± 2%) and SMB (± 11%) with propagation of errors. The polygon and grid setups give very similar spatial patterns to each other. For these two setups, in addition to the measurements errors above, the mass balance estimate is largely affected by errors associated with data interpolations. Because it is difficult to accurately determine the net uncertainties associated with data interpolations, we used two setups (polygon and grid) with distinct data interpolations and consider the differences between them as a guide of representative uncertainties associated with the data interpolations.

We averaged mean mass balance values of all ice columns for each setup and for each $\gamma$ (shown with point symbols in Fig. 7). For a given $\gamma$, the polygon setup give the largest estimate, whereas the grid-setup estimate is smaller by 0.02- 0.03 m a$^{-1}$. Because this difference is smaller than the uncertainty for the flowband-setup estimate (± 0.1 m a$^{-1}$), we argue that the interpolation errors are not dominating these results. Higher $\gamma$ values correspond to lower mass balance, and the sensitivity of mass balance to $\gamma$ is nearly uniform for individual columns.

All the polygons show positive mass balance over the full $\gamma$ range, except for southernmost downstream polygon A3 (the slope-direction codes are shown in Fig. 6a). Along slopes C, E and F, mass balance increases monotonically downstream, whereas mass balance of polygons along A, B and D slopes is more variable. Adjacent southern slopes A and F show contrasting features; the slope A shows the least thickening, whereas the slope F shows the most thickening. For the flowband setup, one column has negative mass balance in the northwest downstream, where the estimated flow divergence is anomalously large.

In conclusion, overall, Blåskimen Island has positive mass balance in the past nine years. Thickening rates vary depending on the setups and the choice of $\gamma$. Over the range of $\gamma$ used here ($0.7 \leq \gamma \leq 0.9$), the mean mass balance varies between +0.25 and +0.37 m a$^{-1}$ for the polygon setup, between +0.21 and +0.35 m a$^{-1}$ for the grid setup, and between +0.12 ± 0.10 and +0.27 ± 0.10 m a$^{-1}$ for the flowband setup. Outside the divide region, gamma tend to be higher (0.8–0.9; Section 4.2); consequently, the thickening rates lean towards the lower end of the estimate above.

## 6  Discussion

### 6.1  Topographic characteristics

According to a recent inventory of ice rises and rumples (Moholdt and Matsuoka, 2015; Matsuoka et al., 2015), Blåskimen Island (651 km$^2$) is larger than 91% of isle-type ice rises (mean: 151 km$^2$). Its summit is 410 m a.s.l., which is higher than 89% of the others (mean: 168 m a.s.l.). Maximum measured ice flow speed (15 m a$^{-1}$) is above the mean of maximum ice flow speed (13 m a$^{-1}$) for isle-type ice rises. Also, the mean bed elevation at -110 m a.s.l. is higher than 86% of the others (-178 m a.s.l.). Overall, Blåskimen Island is one of the larger isle-type ice rises.

Detailed surface DEM developed with the kinematic GPS survey (Fig. 3a) revealed many distinct topographic features including the southwestern ridge and steep slopes in the east. In addition, it confirms that lineations in satellite imagery are associated with surface undulations (Goodwin and Vaughan, 1995). The lineations appear where the surface slope varies largely and most of them are associated with uneven bed topography. This supports the use of satellite imagery as a mean to explore first-order surface and bed topography, when an ice rise remains un-surveyed.

Distinct topographic features of the ice surface revealed with our DEM are not fully represented in continent-wide DEMs (e.g. Bamber et al., 2009; Fretwell et al., 2013, in which spatial resolution is 1 km). Also, the summit heights in those products are 24–40 m lower than our measurements. It remains unclear how much this inaccurate description of topography affects modeling SMB and surface density. Lenaerts et al. (2014) demonstrated that elevated topography associated with ice rises causes orographic precipitations and corresponding precipitation shadow not only over ice rises but also on adjacent ice shelves. Such variations could result in anomalous firn density over the ice shelves, which would result in ill-posed estimates of freeboard thickness and its long-term changes of adjacent ice shelves.

### 6.2  Surface mass balance

We found good agreement in the spatial patterns of stake-measured SMB between 2013 and 2014, and radar-measured SMB between 2005 and 2014 (Figs. 4a and 4b). Relative thicknesses of three layers bounded by radar isochrones (and the surface) vary similarly along the profiles (Fig. 2b), inferring that spatial patterns of SMB have remained similar over this 9 year period, 2005 – 2014. SMB averaged over this period vary largely along the radar profiles between 0.71 m a$^{-1}$ (first quartile) and 0.93 m a$^{-1}$ (third quartile), with its mean of 0.81 m a$^{-1}$.

Large spatial variability in SMB was found on other ice rises as well. King (2004) showed that SMB on the Lydden Ice Rise, Brunt Ice Shelf, is highly variable both at large (tens of kilometers) and small spatial scales (hundreds of meters). They demonstrated that large-scale variations are a result of orographic precipitation, whereas small-scale variations are a result of snow redistribution. On both scales, the contribution from sublimation was found to be relatively small. They highlighted that

SMB is sensitive to surface topography; small variations in surface topography cause small changes in wind speed, which could result in large SMB variations, due to the nonlinear relationship between wind speed and snow transport.

Net impact of these mechanisms on SMB in the DML coast was examined using RACMO2 regional climate model (Lenaerts et al., 2014). Among all the ice rises included in their study, SMB varies by a factor of 2–6 between ice rises. On Blåskimen Island, we found that upwind slopes have 2–3 times the SMB on the downwind slopes, which is within the model prediction

(2–4 times; Lenaerts et al. (2014)). Similar SMB gradients are found over other ice rises in DML: 2–3 times over the Halvfarryggen Ice Dome (Drews et al., 2013) and about 2 times over the Derwael Ice Rise (Drews et al., 2015). We compared the SMB values to local maximum surface slope (not necessarily along the prevailing wind direction) but found no clear relationship between them. We observed numerous sastrugi and harder snow on the northwestern downwind slopes, a clear indication of snow erosion/redistribution processes. Another feature is found near the summit where the surface is virtually flat; here, SMB

is lowered by  10% over  0.5 km. Similar low SMB near the summit has been observed in the Halvfarryggen Ice Dome and the Derwael Ice Rise as well, which were attributed to wind erosion (Drews et al., 2013, 2015).

### 6.3    Present-day mass balance

Amongst all factors that affect the mass balance, ice-flow speed varies most widely (57%), as compared to changes in SMB (17%) and ice thickness (11%) about their mean values. Consequently, the mass balance distribution is more sensitive to the

flow-speed distribution than the other factors. This also explains the low mass balance in the A slope (Fig. 6). Downstream region of the A slope has a lower bed than the central flat basin. As the ice surface is steeper, ice flows faster in this region. Overall, despite the large upwind/downwind contrast in SMB (Fig. 4), differential mass flux compensates for the difference in SMB so that no distinct mass balance patterns were found (Fig. 6). Our mass balance estimates show that Blåskimen Island is thickening almost everywhere, but the thickening rate is smaller near the summit than the flank. If this pattern is persistent for

a long period, it would result in flattening of the ice rise initially. But ice-flow fields would probably be adjusted to the new topography, so the net impact of the ongoing differential thickening on ice topography remains unknown.

### 6.4    Long-term evolution and impact on the adjacent ice shelves

Distinct upward arches in the ice stratigraphy up to ∼40 m depth are caused by low SMB near the summit. We observed upward arches below this depth (Fig. 2c), which are most likely Raymond arches, and are conducting ice-flow modelling experiments

to interpret them. Regardless of its cause, the upward arch locations can be a proxy of the summit position in the past (Nereson et al., 1998).

Since the arches in the top ∼40 m are aligned vertically below the summit (Fig. 2b), it indicates that the summit position has been stable over the past several decades. In contrast, the deeper arches (∼300 m a.s.l. and below) show more offset towards

the southeast as it goes deeper (radar profile 2–2' shown in Fig. 2c). This trend is also found in other radar profiles. A possible interpretation of this arch inclination is that the summit has migrated towards northwest in the past. We do not see clear signs of present-day mass imbalance implying such divide migration (Fig. 6). It may indicate recent changes in mass balance and/or a limitation of our mass balance estimate due to a coarse resolution (thousands of meters), compared to the observed arch offset (100–400 m).

If the upward arches at greater depths are Raymond arches, it indicates that the summit position has been stable within several ice thicknesses from the current summit over one characteristic time $T$ ($= H/SMB$, $\sim$610 years at Blåskimen Island) or longer. If the summit position stays stable for longer time, then Raymond arches are further developed into double-peaked arches (Martín et al., 2009a), which are not clearly observed here (small side arch is caused by bed bump nearby, according to our initial ice-flow modeling). According to Martín et al. (2009a), the double-peaked arches appear after several $T$, though its mature shape is reached only after $\sim$10$T$ or so. Therefore, we speculate that the summit of Blåskimen Island has been stable within several kilometers at least in the past $\sim$600 years but no longer than several millennia.

Roles of ice rises vary largely in terms of its settings, and thus can change during the evolution (Matsuoka et al., 2015). Blåskimen Island is currently situated at the calving front of the local ice shelves and in the ice-flow shadow of Novyy Island ice rise upstream (Fig. 1). This setting implies that currently Blåskimen Island alone has limited impact on the continental grounding line and ice flux from the ice sheet. However, it seems likely that Blåskimen Island plays a more significant role than the Novyy Island to maintain the current calving front position. Favier and Pattyn (2015) demonstrated that an ice shelf landward of the ice rise is thicker than the seaward ice shelf facilitating formation of rifts and ice-shelf breakups just seaward of the ice rise. To explore the dynamics of this surrounding region better we will use the datasets presented in this study to model the evolution of the ice rise.

## 7    Conclusions

Ice rises are a useful resource to investigate evolution and past climate of the DML coastal region. We investigate Blåskimen Island ice rise, one of the larger isle-type ice rises at the calving front of the intersection of Fimbul and Jelbart Ice Shelves, using geophysical methods. It has an overall dome shape and at its summit it is $\sim$350 m above the adjacent ice shelf. It stands over a flat bed with a mean elevation of 110 m below the sea level. The ice flows from the summit towards the flank with speeds up to 15 m a$^{-1}$. We found a good agreement in the spatial patterns of stake-measured Surface Mass Balance (SMB) between 2013 and 2014, and radar-measured SMB between 2005 and 2014. Both show higher SMB on the upwind slopes (southeast), in comparison to downwind slope (northwest) by $\sim$37%. This variation is likely a result of orographic precipitation during storms. Using the Input-Output method for a range of parameters and column setups, we conclude that Blåskimen Island has been thickening over the past decade. Thickening rates cannot be determined precisely, but ensemble results show that thickening rate averaged over the ice rise can be between 0.12 m a$^{-1}$ and 0.37 m a$^{-1}$. On longer timescales, we speculate that the summit of Blåskimen Island has been stable within several kilometers at least in the past $\sim$600 years but no longer than several millennia.

## 8 Data availability

Field data (GPR, GPS, borehole thermistor data) and the derived datasets (ice thickness, flow speed, and surface mass balance) will be released at: http://data.npolar.no on the completion of the review process. They are available for the editor and reviewers upon request. The 23 m long firn-core data and their availability are described in Vega et al. (2016).

*Author contributions.* Matsuoka and Brown designed the study. All three authors conducted fieldwork. Goel led the overall data analysis and interpretations. Brown prepared the GPS and GPR processing workflow, which Goel adapted. Brown also produced inversion SMB estimates. Goel and Matsuoka prepared the manuscript, and Brown contributed to finalize it.

*Competing interests.* Author KM is a member of the editorial board of the journal.

*Acknowledgements.* This work was funded by grants from Norwegian Antarctic Research Expeditions (NARE) and the Center for Ice Climate and Ecosystems (ICE) of the Norwegian Polar Institute. Vikram Goel received a PhD studentship from National Centre for Antarctic and Ocean Research (NCAOR), through a financial support from the Ministry of Earth Sciences (MoES), Government of India. We thank Harvey Goodwin, Kjetil Bakkland, Ørjan Karlsen, and Peter Leopold for their contributions to fieldwork. Troll Station of NARE and SANAE Station of the South African National Antarctic Programs provided field support. Tor Ivan Karlsen developed the low-frequency radar. Carmen P. Vega, Elisabeth Isaksson, and their team provided the depth-age data of the core. We thank our two reviewers for their constructive comments, which improved the manuscript significantly. Figure 1 was developed using a free GIS data package Quantarctica (quantarctica.npolar.no).

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

# Figures

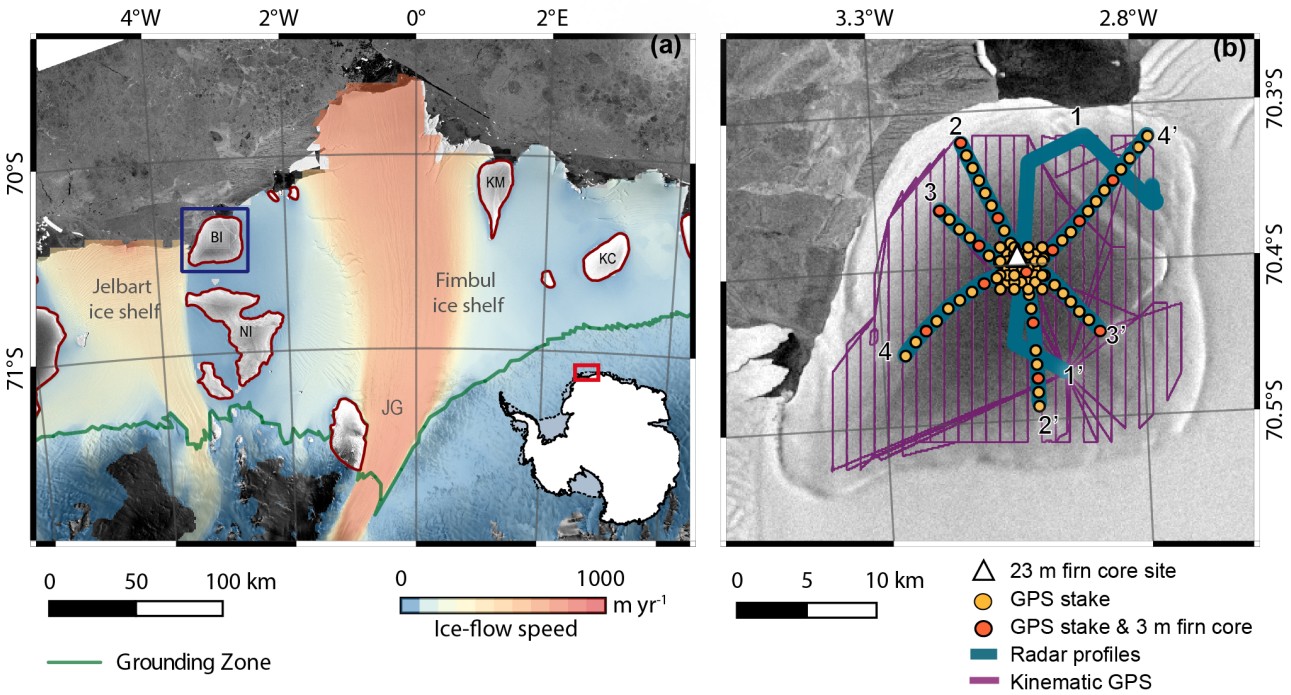

**Figure 1.** Blåskimen Island ice rise, Western DML. (a) Blaskimen Island (squared) located between the Fimbul and Jelvart ice shelves. Inset shows the coverage of this map. Ice rises are outlined in red (Moholdt and Matsuoka, 2015), and the grounding zone of the ice sheet is illustrated in green (Bindschadler et al., 2011). Color shows flow speed of the ice sheet and ice shelf (Rignot et al., 2011). The background of the both panels is Radarsat-1 satellite imagery (Jezek et al., 2002). Acronyms stand for BI: Blåskimen Island, JG: Jutulstraumen Glacier, KM: Kupol Moskovskij, KC: Kupol Ciolkovskogo and NI: Novyy Island. (b) Close-up view of Blåskimen Island. Pink curves show kinematic GPS profiles used to determine the surface topography and green curves show radar profiles. Yellow circles show GPS stake positions for ice-flow measurements. Red circles show GPS stake position where 3 m long firn core was also drilled. The white triangle shows the location where the 23 m long firn core was drilled. Maps are projected to the Antarctic Polar Stereographic view (EPSG3031).

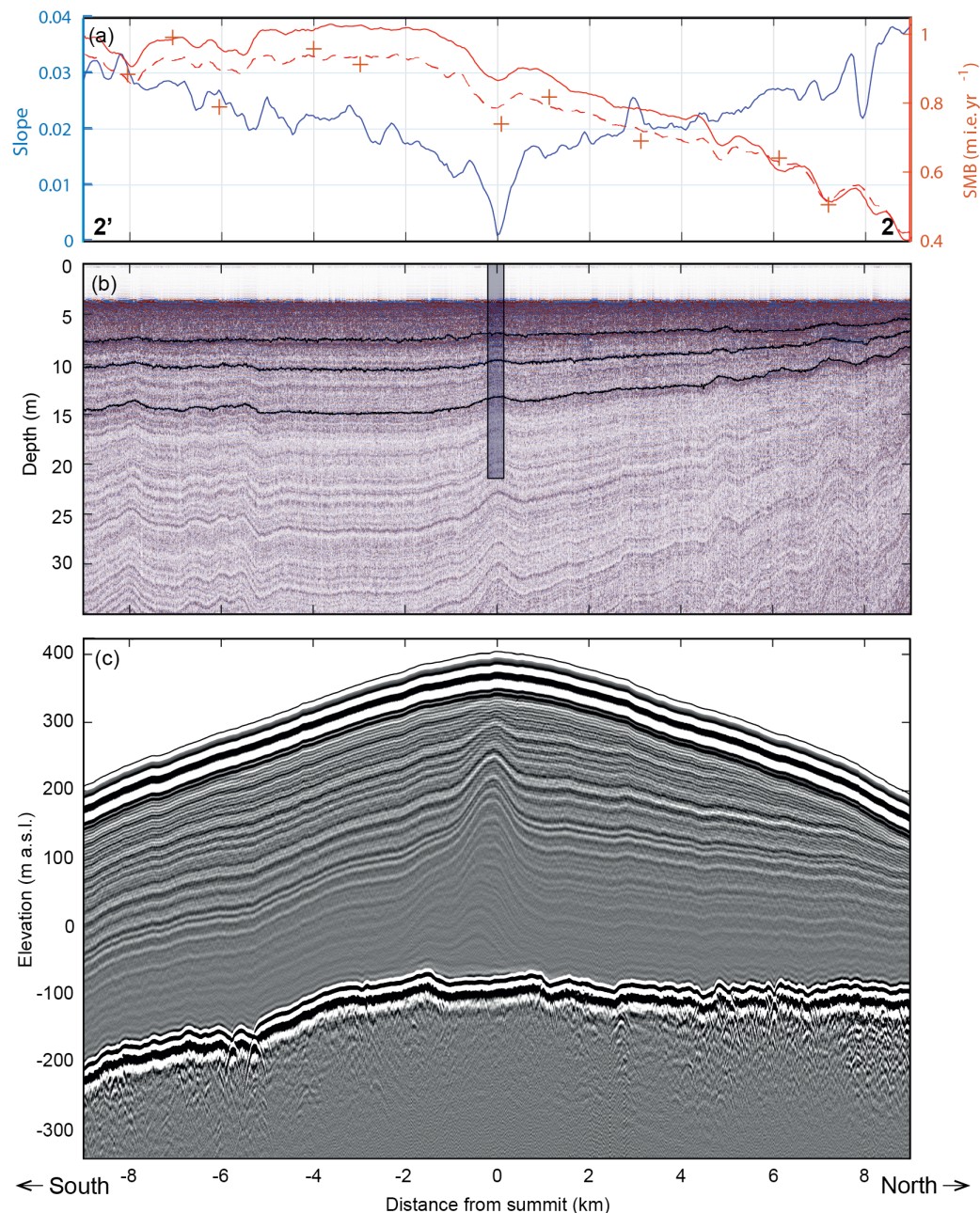

**Figure 2.** Cross sections of the ice rise along the flowline 2–2' (Fig. 1b). (a) Surface slope (absolute number, left axis) and SMB (right axis). '+' markers show SMB derived from stake heights. Red curves show the two SMB estimates derived from radar data, with solid curve assuming only vertical variability in density, and dashed curve accounting for both vertical and lateral variability in density. (b) 400-MHz radargram. Three englacial reflectors are highlighted, which are dated with the 23 m long firn core (vertical bar) and used to determine SMB. (c) 2-MHz radargram. Data are shifted using the GPS-measured surface elevations to show the topography.

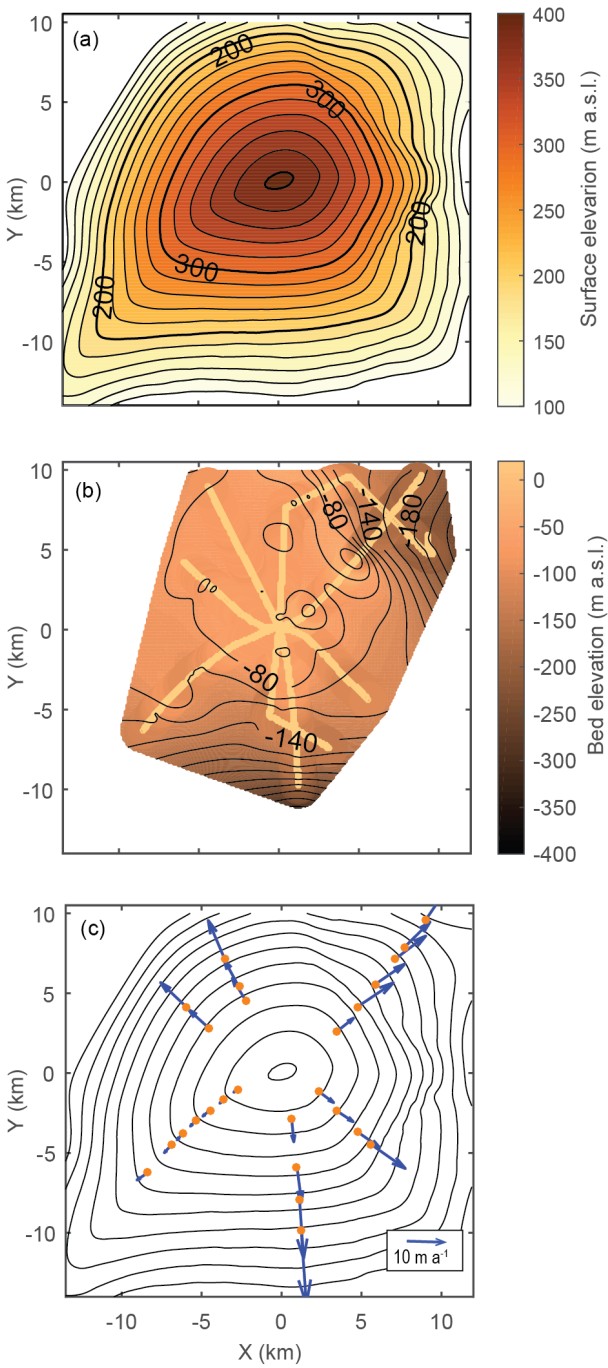

**Figure 3.** Ice surface (a), bed topography (b) and ice-flow field (c) of the ice rise. The local coordinates are parallel to the polar stereographic coordinates EPSG3031 and centered at the ice-rise summit. (a, b) Elevation contours are 20 m intervals. In panel b, radar profiles used to derive the bed topography are highlighted in yellow to show the data availability. Kinematic GPS survey locations are shown in Fig. 1b. (c) Surface ice-flow velocities (blue arrows originated from orange circles) overlaid on surface topography (30 m interval contours). There are more GPS stakes near the summit (Fig. 1b), which did not give significant results.

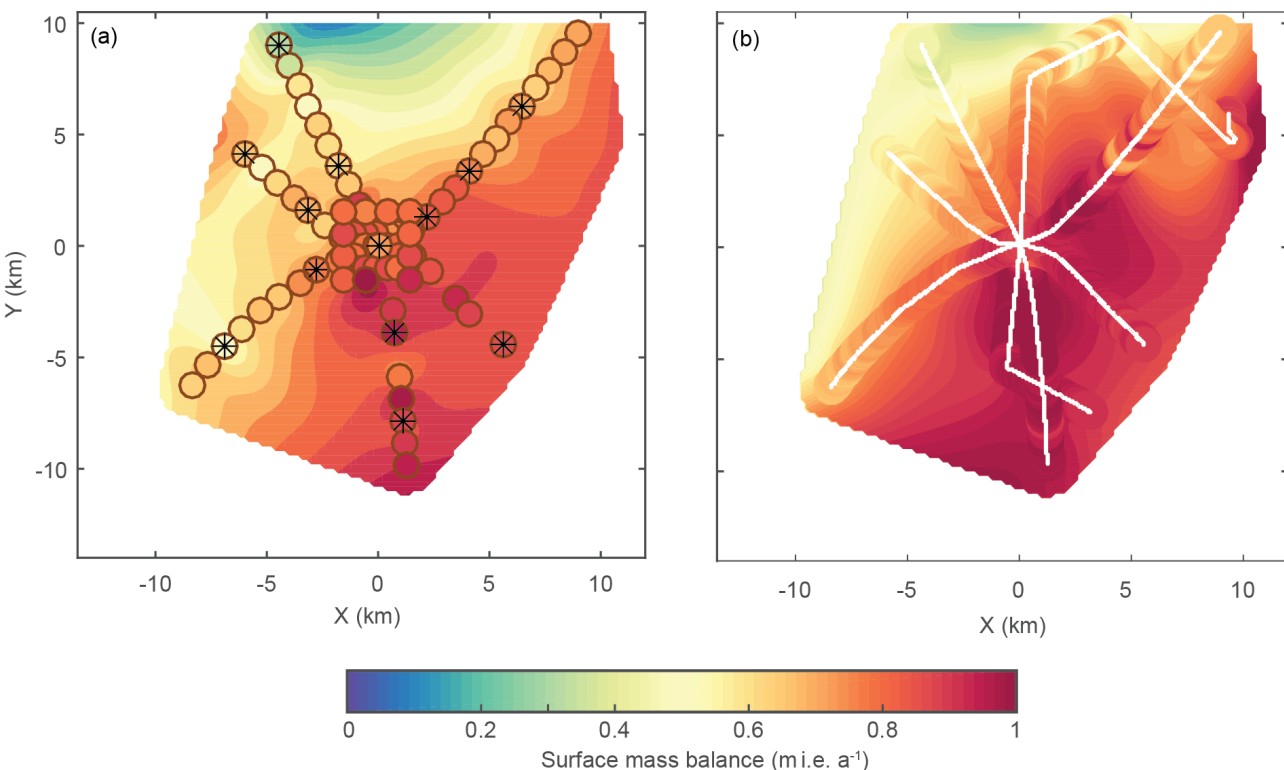

**Figure 4.** Surface mass balance (SMB). (a) SMB estimated with stake methods over the 2013-14. Circles show the locations of the installed stakes. Wheel spokes inside of the circles show the locations of 13 surface-density measurements. (b) SMB estimated with radar over the past decade, by accounting only the vertical variability of density. White curves show radar profiles. The 23 m long firn core was drilled near the crossover of all profiles at the summit.

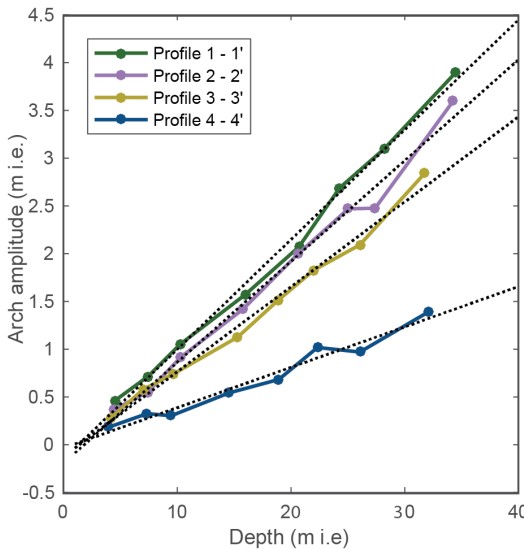

**Figure 5.** Depth variations of upward-arch amplitudes observed near the summit (Fig. 2b) along four radar profiles (Fig. 1b). Dots show the arch amplitudes and dashed lines show the linear fits of the arch amplitudes to depth.

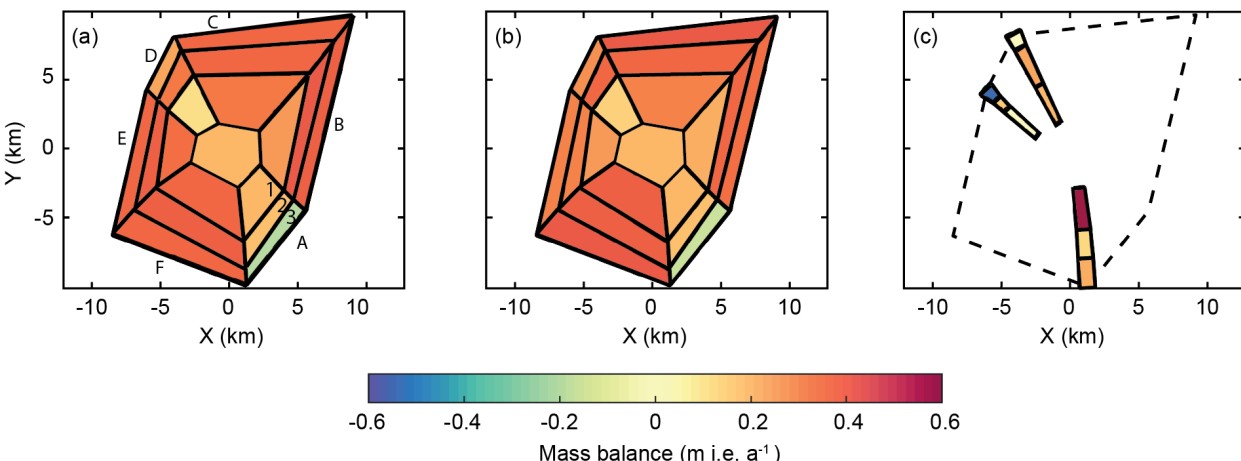

**Figure 6.** Recent mass balance derived with three different setups when the fraction $\gamma$ of depth-averaged flow speed to the surface flow speed is equal to 0.8. (a) Polygon setup. The ice rise is divided into 6 slopes (A–F) with 3 polygons (1–3, 1 for most upstream in each slope) and a summit polygon. (b) Grid setup. (c) Flowband setup. The dashed lines show the extent of polygon and grid setups for comparison.

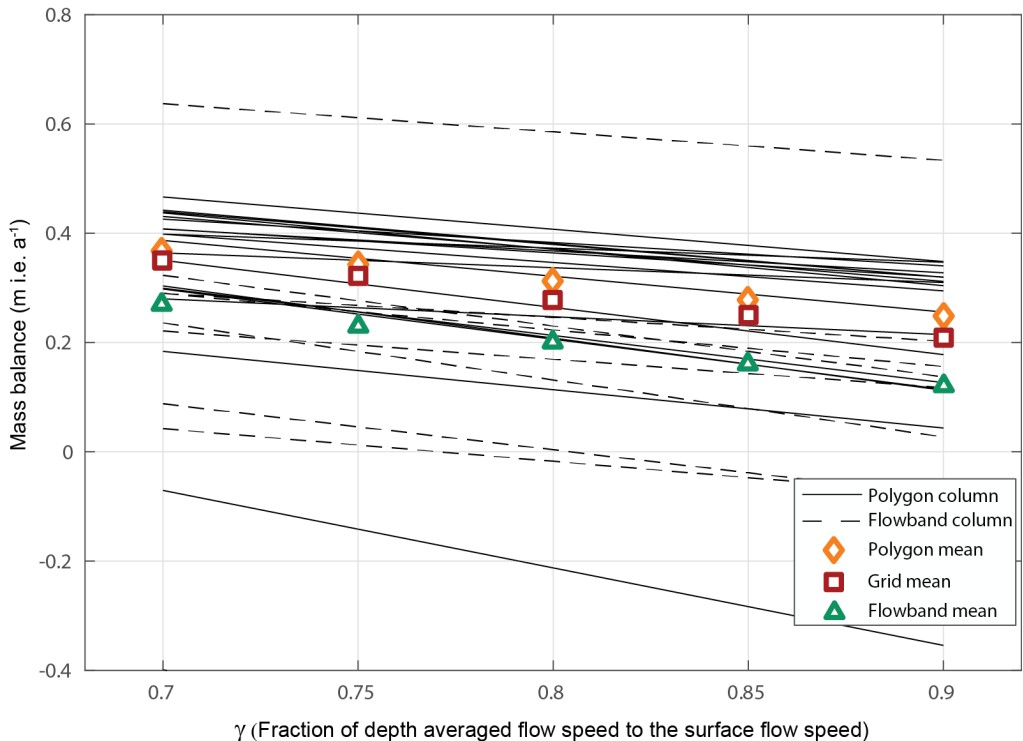

**Figure 7.** Mass balance estimates of individual polygonal column (Polygon setup, solid lines) and flowband columns (Flowband setup, dashed lines) in terms of $\gamma$, the fraction of the depth averaged flow speed to the surface flow speed. Those derived with the grid setup (not shown) are very similar as those with the polygon setup. The ensemble-mean mass balance estimates for each $\gamma$ are shown for all the setups.