# Peer review of "Glaciological settings and recent mass balance of Blåskimen Island in Dronning Maud Land, Antarctica"

_The Cryosphere, 2017_

## Referee Comment (RC1) · Anonymous Referee #1 · 9 Jun 2017

**Review of "Glaciological settings and recent mass balance of the Blåskimen Island in Dronning Maud Land."**

**General Comments**

Goel and coauthors report field observations from a relatively large ice rise that sits between Jelbert and Fimbul Ice Shelves in Dronning Maud Land. They use two ice-penetrating radars with different frequencies, stake measurements and surface GPS, to constrain the topography, thickness and internal stratigraphy of the ice rise as well as obtain several spatially-distributed estimates of mass balance.

The observations are well reported and are likely to be of interest to the glaciological community. Specifically, they may interest those interested in the surface mass balance and long-term evolution of the ice rises in Dronning Maud Land. As the authors mention, ice rises are important because they influence regional ice dynamics and encode information about the stability or otherwise of these dynamics in their internal stratigraphy. Ice rises also influence atmospheric circulation and therefore surface mass balance and are useful locations for drilling ice cores. Overall, any information we can gain on the glaciological setting and mass balance of ice rises, as well as their long-term stability is useful and worth publishing. In my opinion, this work should be published in the Cryosphere after revision to address the comments below.

Overall, the manuscript is written well, with some exception. There are many grammatical and other errors, as well as some passages that I found it difficult to understand. These difficult-to-understand phrases, various errors and other places where I propose that rewording could improve the paper, are indicated below in "Technical Corrections". Note that due to the high density of errors, there are likely to be many that I missed and I suggest that any revised manuscript is read carefully for this kind of error.

Individual scientific comments are described below in the "Specific Comments" section.

**Specific Comments**

Section 5.2 describes how shallow radar layers are used to derive SMB. This involves multiplying the depths of three dated layers by the snow/firn density to compute the mass accumulated since each layer was laid down. Spatial variations in density are taken account of using two methods. Very little detail is provided of the second method. An in preparation manuscript is cited and one sentence (L3-5, P7) describes the inverse method. Because (1) very little detail is provided (certainly too little to be able to reproduce the results) and (2) this method is apparently used to take account of a spatial variation in near-surface density of only $\pm2\%$, I suggest this alternative method is removed from the manuscript. The first method takes account of the much larger vertical variation in density and seems adequate. I would suggest removing the second method and simply stating that in this case the $\pm2\%$ variation in density is small and that it is possible to alternatively take account of this at the expense of taking into account the vertical variations and that a manuscript is in preparation that will report ongoing work related to doing this with an inverse method.

Page 5, L14-15: A value for the SMB of the whole island is quoted and an uncertainty is estimated based on measurement errors and uncertainty accounting for snow densification. Does this uncertainty estimate also take into account uncertainty associated with the interpolation? This is difficult (as the authors point out later in the manuscript), so if not, perhaps it is better to quote the SMB as the average of the stakes measurements, rather than as the average for the whole island, as this reads to me.

Conclusions, page 13, L21-22: As sastrugi patterns can change from one day to the next, I am not sure that this a robust conclusion to draw about SMB patterns from unquantified observations of sastrugi in the

field. I agree that in general it is tempting to conclude that there must be lower accumulation where you see sastrugi in the field (I have seen this myself on lower accumulation sides of ice rises), so I think it is reasonable to mention this earlier in the manuscript, but I don't think it belongs here as one of your main conclusions.

**Technical Comments**

Title: To my reading the 'the' before 'Blåskimen Island' is unnecessary. Indeed, in the abstract the name of the ice rise appears without the 'the'.

*Page 1*

L7: Suggest replacing the sentence starting with "Radar stratigraphy…" with "Arches in radar stratigraphy observed with radar suggest that the summit of the ice rise has been stable for ~600 km." This avoids introducing the concept of characteristic time in the abstract, which may be confusing to those not familiar with it.

L11: Is there a reference for the statement about 74% of the coastline being surrounded by ice shelves?

L14: 'are' → 'is'.

L15: Suggest delete 'eventual'.

L16: 'the ice shelf' → 'ice shelves'.

L16-17: this sentence about buttressing seems to repeat what was said in the previous paragraph.

L23: Two more recent papers that use isochrones to constrain millennial scale ice-rise evolution that could be referenced here are:

Drews, R., Matsuoka, K., Martín, C., Callens, D., Bergeot, N. and Pattyn, F., 2015. Evolution of Derwael ice rise in Dronning Maud Land, Antarctica, over the last millennia. *Journal of Geophysical Research: Earth Surface*, *120*(3), pp.564-579.

and

Kingslake, J., Martín, C., Arthern, R.J., Corr, H.F. and King, E.C., 2016. Ice- flow reorganization in West Antarctica 2.5 kyr ago dated using radar- derived englacial flow velocities. *Geophysical Research Letters*, *43*(17), pp.9103-9112.

*Page 2*

L1: Suggest replace 'using an ice core from the summit' with 'using ice cores drilled through ice rises'.

L4: insert comma after '(Fig. 1a)'.

L12: Is it narrower by a factor of two and slower by a factor of two or does the factor of two only apply to the 'slower'? Suggest switch order to '…slower by a factor of two and narrower…' if appropriate, to avoid confusion.

L16: Reword the last sentence. 'Whereas towards east…' doesn't read well to me.

L19: 'summer' → 'summers'.

*Page 3*

L3: 'were' → 'was'

L4: How many were lost and why were they lost? Were they buried? Also did you have a criterion for rejecting stakes which were tilted too much? Surely all stakes are tilted slightly. If so, it's not correct to say here that you rejected stakes that were tilted.

L5: Suggest move 'relative' to between 'stakes' and 'to'.

L6: Suggest 'infer approximate' → 'estimate'.

L13: Suggest'to the sea level' → 'to heights above local sea level'.

L13: This reads as if you used the gravity product to subtract the value. Suggest reword to say that you subtracted the value that was supplied by the gravity product.

L19: 'profiles' → 'surveys'. It looks like you did more than two profiles.

L21: Delete 'a'.

L22: Delete commas after 'radar'.

L23: Insert 'a' before dewow and Ormsby. Also 'Post processing was made…' does not read well.

L24: 'the' →'a' and insert 'of' after 'speed'.

L25: How is the firn correction computed? Using the firn cores presumably, but this is not mentioned here.

L29: Transported the samples where?

L30: Suggest delete 'backed'.

*Page 4*

L1: 'to' → 'in', insert 'the' between 'from' and 'surface'.

L5: delete 'of'.

L9-10: suggest delete 'giving a relative dome-shaped topography to the ice rise'.

L11: Not clear what 'a flatter basin northeast' means and it is grammatically incorrect. I cannot see a clear distinction between slopes in the northeast and in the other directions. Also I can't see clear distinctions between basins in figure 2a.

L12: The line 2-2' isn't strictly a flow line as it passes through the ice divide. I guess it is two flow lines connected together, but describing it as a flow line I don't think is correct.

L17: The estimate of the vertical uncertainty of ±5m could be more fully explained here. The center frequency of the deep radar was 2 MHz, corresponding (I think) to a wavelength of $c/n_i/2x10^6 = 84.2$ m, where c is the speed of light and $n_i$ is the refractive index of ice. This wavelength is considerably more than your estimated uncertainty in digitizing the bed reflector. Is the higher precision achieved due to the signal being quite broadband? Perhaps this can be explained in more detail, as one might expect a bed reflector imaged with an 84 m wavelength radar to manifest as a layer thicker than 5 m.

L27: Suggest delete 'below'.

L28: The surface velocity measurements are described as a surface velocity field here, when they really appear as just point measurements in figure 2c, rather than a field.

L28: Suggest 'from' → 'of'.

*Page 5*

L9: Is this surface density or the density averaged over the top 3 m of all the cores? It is confusing because the previous sentence mentions "surface density variations", but this sentence only says 'firn density'.

L10: Suggest replace 'To estimate..' with 'When estimating SMB below…', or similar.

L17: It is stated here that *most* SMB values lie between the first and third quartiles, but doesn't this by definition mean that half the measurements lie between these two values (i.e. half ≠ most)?

L22: I am not sure I see the relevance here of the parentheses about wind direction.

*Page 6*

L6: Is it not more precise to say that it is the larger vertical strain rates near the surface at the divide (rather than strong horizontal variations in vertical strain rates) due to the Raymond effect that could mean that the shallow-layer approximation may not be valid at the divide. Gillet-Chaulet et al. (2013) and Kingslake et al. (2014) and probably others have measured higher magnitude vertical strain rates near the surface on ice divides and could be cited here to support the point.

Gillet‐ Chaulet, F., Hindmarsh, R.C., Corr, H.F., King, E.C. and Jenkins, A., 2011. In‐ situquantification of ice rheology and direct measurement of the Raymond Effect at Summit, Greenland using a phase-sensitive radar. *Geophysical Research Letters*, *38*(24).

Kingslake, J., Hindmarsh, R.C., Aðalgeirsdóttir, G., Conway, H., Corr, H.F., Gillet‐ Chaulet, F., Martin, C., King, E.C., Mulvaney, R. and Pritchard, H.D., 2014. Full‐ depth englacial vertical ice sheet velocities measured using phase‐ sensitive radar. Journal of Geophysical Research: Earth Surface, 119(12), pp.2604-2618.

Eqn 2: This equation appears incorrect. The $z$ on the top of the fraction in the integrand on the right side shouldn't be there.

L30: I do not see why using a constant density (when in fact the density varies with space) would make the problem ill-posed. It may make the MB inaccurate, but why ill-posed?

L32: Is there a word missing from near the start of this sentence?

*Page 7*

L19-20: There is a comparison made here between SMB derived with two different methods. In my specific comments above I suggest that the second of these methods be removed from the manuscript, but here I would suggest that if both methods must be included that the comparison is made clearer here. As figure 4 stands, we have to compare spatial patterns in the two fields, but much of the structure in these fields appears to depends on the details of the interpolation, i.e. how the surface is interpolated across the large gaps between data points. The statement here that the two methods 'give nearly identical spatial patterns in SMB' doesn't appear to be correct, but the differences that stand out in figure 4 are due to the interpolations. So, I suggest that you include a plot that compares the two datasets without including an

interpolated field. For example, a profile along the line 2-2' of the SMB estimates, or remove the spatial dimension entirely and include a scatter plot the estimates from each method on each axis.

L30-31: There is a missing link here. Is the point that if there were basal melting then the Raymond arches would be smaller? As the arches heights have not been compared to arch heights expected from modelled (Martin et al., 2006) or measured (Kingslake et al., 2016) vertical velocities, I am not sure that their size can be used to support the statement about melting. Note that I think it is fine to assume that the ice rise is cold based if the 1D modelling shows that it is thin enough.

Kingslake, J., Martín, C., Arthern, R.J., Corr, H.F. and King, E.C., 2016. Ice‑ flow reorganization in West Antarctica 2.5 kyr ago dated using radar‑ derived englacial flow velocities. *Geophysical Research Letters*, *43*(17), pp.9103-9112.

*Page 8*

The term laminar flow is used at least three times here to distinguish the flow of the ice rise flanks from flow within a region close to the divide where the Raymond effect acts. All glacial ice flow is laminar (as opposed to turbulent), so this seems doesn't seem like the correct term to use. Perhaps a better way to make the distinction is to say that in the flanks the shallow-ice approximation is valid whereas in the divide region it is not. The approximation of γ in line 7 comes from the SIA and Martin et al. (2009) showed that the full-stokes models are required to describe the Raymond effect and discuss how the SIA is incapable of this. So perhaps this works better.

Martin, C., Hindmarsh, R.C. and Navarro, F.J., 2009. On the effects of divide migration, along‑ ridge flow, and basal sliding on isochrones near an ice divide. Journal of Geophysical Research: Earth Surface, 114(F2).

L26-27: Can you expand on the statement that a more accurate determination of γ requires us to know ice-flow history? Is this because ice rheology has a memory through ice temperature and crystal fabric?

*Page 9*

L1-2: This sentence, starting 'This is because… ' is very unclear.

I found section 6.2.3 very difficult to understand. For example, in line 21, is 'this estimate' the estimate of SMB using the flowband setup or the estimate of the variation in flowband width. Also is the second paragraph (which is just one sentence) missing more material? I suggest that the description of the flowband setup is re-written and expanded to make this clearer.

L6: I am confused by this statement that for a given γ the polygon setup gives the largest estimate of mass balance. According to Fig 7, when γ > 0.75 the flowband setup (dashed line) is higher than the polygon setup (solid line).

L7: Suggest remove 'much'. I am not sure that I would describe 0.05-0.07 m/a as *much* smaller than 0.1 in this context. They are the same order of magnitude.

L8-9: I am not sure that I understand the sentence starting 'Higher γ….'. Are you saying that the dependence of the mass balance on γ is nearly linear. If so, I suggest you replace 'uniform' with 'linear'.

L11: Delete 'as it goes'.

L12: Rephrase this sentence to avoid the phrase 'varies in a more variable way'.

*Page 11*

The discussion contains many typos and ungrammatical sentences, that I have not listed in detail.

L15-26: Do these two paragraphs belong in the introduction? They do not discuss any of the new results and I think they can be shortened without losing substance.

*Page 12*

L20: Another recent relevant reference is:

Kingslake, J., Martín, C., Arthern, R.J., Corr, H.F. and King, E.C., 2016. Ice- flow reorganization in West Antarctica 2.5 kyr ago dated using radar- derived englacial flow velocities. *Geophysical Research Letters*, *43*(17), pp.9103-9112.

L26: 'inferring' → 'implying'.

L27: '(1000s m)' is confusing as throughout the manuscript spaces between values and units have not been consistent. Suggest replace with '1000's of meters'.

L29: Delete 'the' before 'Raymond'.

L31: Is a word missing from near the beginning of this sentence?

*Page 13*

Section 7.5 contains no discussion of your results, only the location of the ice rise. Can any of your findings (for example bed topography) contribute to this discussion of the impact of the ice rise on the adjacent ice shelves? If not, perhaps this belongs in the introduction or the section in which you introduce the ice rise.

L17: Insert 'a' between 'over' and 'flat'.

*Figures and captions*

Figure 4 caption: The second sentence appears to be missing something.

Figure 5 caption: The end of second sentence is ungrammatical.

Figure 7 caption, lines 3: Replace 'as' with 'to'.

---

## Referee Comment (RC2) · Anonymous Referee #2 · 4 Jul 2017

This manuscript provides exhaustive details on the field measurements and data analysis over an ice rise in Dronning Maud Land, Antarctica from radar and GPS. While the manuscript provides sufficient data to support their conclusions it falls short of providing a compelling story. I think the main problem may be more due to writing however, than mis-interpretation of the data. Indeed, it seems to be more of a field report than a scientific paper that tells a story. I would prefer for the authors to follow the more traditional style of writing with sections called: introduction, data and methods, results, discussion, conclusions. While these sections are in the text, there are numerous other sections and subsections with detailed headings making it difficult to follow the logical order of the manuscript. Further, the writing could be tightened by combining very short

paragraphs and reducing redundant information. For example, instead of presenting info on the ice core records in section 3, simply refer to the reference in your discussion of your measured SMB. Same for temperature info - why is this relevant to the current study? This kind of thing led to my inability to understand what the story is here.

Page 3, Line 21: delete "a" Page 3, Line 30: If you're using the firn cores to get estimates of density variability it might be nice to include a figure of these data. Page 5, Line 8: This sentence is not needed as it was already said in a previous section. Page 6, Line 6-7: Don't forget the work of Nereson and Raymond, 2001 Page 7, Line 7: delete "a" Page 7, Line 25: Rignot and Kanagaratnam is a better reference since they (I believe) pioneered the IO method. Page 7, Line 28: Not sure you can assume no melting based on the study by Neumann et al., (2008) which is on the other side of Antarctica. Page 11, Lines 3-9: The authors make an offhand assertion that current DEMs of Antarctica do not properly resolve this ice rise elevation, nor the details contained within their DEM. I think they would have to demonstrate that the added detail is necessary in order to "get the precip right" according to Lenaerts (2014). I might be convinced that gross differences between DEMs are important but I'm not convinced that we need to include every small detail as obtained by a ground survey. Page 11, Line 31-32: your reported slope differences are within the expected range of the model prediction by Lenaerts et al, (2014) not "smaller than" Page 12, Line 19: I don't see the need for these references here - we already know what Raymond arches are. Page 12, Line 21: refer to Nereson's work on this Page 12, Line 30-32: I can clearly see double-peaked arches in Fig. 3c.

---

## Editor Comment (EC1) · O Eisen (Editor) · 1 Aug 2017

Dear authors

having received two reviews it is your turn now to provide a substantial revision if your manuscript should further be considered for publication in TC. The revision will undergo a second round of reviews, on which the final decision will be based.

Regards Olaf
* * *

---

## Author Comment (AC1) · 23 Aug 2017

Please find our detailed response to the reviewers comments in the attached document.

Please also note the supplement to this comment:
https://www.the-cryosphere-discuss.net/tc-2017-61/tc-2017-61-AC1-supplement.pdf
* * *

---

## Author Response (AR1)

**Response Letter**

We appreciate two reviewers for investing their time and providing constructive comments on our manuscript. Below, we explain changes we made and present our opinions to the points that we did not follow. We hope that these revisions are satisfactory, and the revised manuscript would meet journal's criteria.

Some major changes to the manuscript are –

- The introduction is reorganized/rephrased to emphasize more on the motivations behind the study.
- The manuscript structure is reorganized to a more conventional structure as recommended by reviewer #2. The charges in the structure are shown the in figure below –

**Previous**
1. Introduction
2. Blåskimen Island
3. Field measurements
    1. Kinematic and static GPS surveys
    2. Ice- penetrating radar profiling
    3. Firn cores and borehole-temperature measurements
4. Topography and surface flow field
5. Surface mass balance
    1. SMB derived from stake heights
    2. SMB derived from shallow-sounding radar stratigraphy
6. Mass balance of the ice rise
    1. Methods
    2. Estimate setups
        i. Polygon setup
        ii. Grid setup
        iii. Flowband setup
    3. Uncertainties
    4. Results
7. Discussion
    1. Topographic characteristics
    2. Surface mass balance
    3. Present day mass balance
    4. Long-term evolution
    5. Impact on the adjacent ice shelves
8. Conclusions

**New**
1. Introduction
2. Blåskimen Island
3. Field measurements and data processing
    1. Kinematic and static GPS surveys
    2. Ice-penetrating radar profiling
    3. Firn cores and borehole-temperature measurements
4. Analytical methods
    1. Surface mass balance
        1. Deriving SMB using stake heights
        2. Deriving SMB using shallow-sounding radar stratigraphy
    2. Mass balance of the ice rise
        1. Constraining γ
        2. Estimate setups
            1. Polygon setup
            2. Grid setup
            3. Flowband setup
5. Results
    1. Topography and surface flow field
    2. Surface mass balance
    3. Mass balance
6. Discussion
    1. Topographic characteristics
    2. Surface mass balance
    3. Present day mass balance
    4. Long-term evolution and impact on the adjacent ice shelves
7. Conclusions

- Based on recommendation from reviewer #1, we have used the first radar method to derive SMB accounting for the vertical variability in density (not lateral variability across the whole ice rise). Consequently, the mass balance estimate is updated based on this SMB as input. Figure 2, 4, 6 and 7 were updated with the new estimate.

Below, we quote reviewers comments first, and then our responses in blue italic fonts follow. Otherwise stated, page/line numbers in this letter refer the numbers in the marked manuscript. Due to a technical issue with the software we use (Latexdiff) to prepare the marked manuscript, changes in section names are not highlighted in the marked manuscript. Also, long citations overflow the page on a few occasions. We have listed these citations at the end of this response letter.

**Responses to Reviewer #1**

**Review of "Glaciological settings and recent mass balance of the Blåskimen Island in Dronning Maud Land."**

**General Comments**

Goel and coauthors report field observations from a relatively large ice rise that sits between Jelbert and Fimbul Ice Shelves in Dronning Maud Land. They use two ice-penetrating radars with different frequencies, stake measurements and surface GPS, to constrain the topography, thickness and internal stratigraphy of the ice rise as well as obtain several spatially-distributed estimates of mass balance.

The observations are well reported and are likely to be of interest to the glaciological community. Specifically, they may interest those interested in the surface mass balance and long-term evolution of the ice rises in Dronning Maud Land. As the authors mention, ice rises are important because they influence regional ice dynamics and encode information about the stability or otherwise of these dynamics in their internal stratigraphy. Ice rises also influence atmospheric circulation and therefore surface mass balance and are useful locations for drilling ice cores. Overall, any information we can gain on the glaciological setting and mass balance of ice rises, as well as their long-term stability is useful and worth publishing. In my opinion, this work should be published in the Cryosphere after revision to address the comments below.

Overall, the manuscript is written well, with some exception. There are many grammatical and other errors, as well as some passages that I found it difficult to understand. These difficult-to-understand phrases, various errors and other places where I propose that rewording could improve the paper, are indicated below in "Technical Corrections". Note that due to the high density of errors, there are likely to be many that I missed and I suggest that any revised manuscript is read carefully for this kind of error.

> *We thank all of your efforts. We regret that the previous manuscript includes typos and grammatical errors, which we tried to fix as much as we can in this version.*

Individual scientific comments are described below in the "Specific Comments" section.

**Specific Comments**

Section 5.2 describes how shallow radar layers are used to derive SMB. This involves multiplying the depths of three dated layers by the snow/firn density to compute the mass accumulated since each layer was laid down. Spatial variations in density are taken account of using two methods. Very little detail is

provided of the second method. An in preparation manuscript is cited and one sentence (L3-5, P7) describes the inverse method. Because (1) very little detail is provided (certainly too little to be able to reproduce the results) and (2) this method is apparently used to take account of a spatial variation in nearsurface density of only ±2%, I suggest this alternative method is removed from the manuscript. The first method takes account of the much larger vertical variation in density and seems adequate. I would suggest removing the second method and simply stating that in this case the ±2% variation in density is small and that it is possible to alternatively take account of this at the expense of taking into account the vertical variations and that a manuscript is in preparation that will report ongoing work related to doing this with an inverse method.

> *We agree with the reviewer and use the SMB estimated with the first radar method to derive mass balance of the ice rise. However, we briefly mention the second radar method and results obtained with this method to give an approximate idea on errors in SMB estimates related to the uncertainties of density.*

Page 5, L14-15: A value for the SMB of the whole island is quoted and an uncertainty is estimated based on measurement errors and uncertainty accounting for snow densification. Does this uncertainty estimate also take into account uncertainty associated with the interpolation? This is difficult (as the authors point out later in the manuscript), so if not, perhaps it is better to quote the SMB as the average of the stakes measurements, rather than as the average for the whole island, as this reads to me.

> *No, uncertainty associated with the interpolation is not included in the error estimate. As the reviewer said, it is difficult to do. In the revised manuscript, the mean SMB only used the stake measurements. We have rephrased the text at P12L2 to make it clear.*
>
> *"The mean SMB from 90 stake-height measurements across Blåskimen Island is 0.78 m a$^{-1}$, for the period between January 2013 and January 2014 (Fig. 5a)."*

Conclusions, page 13, L21-22: As sastrugi patterns can change from one day to the next, I am not sure that this a robust conclusion to draw about SMB patterns from unquantified observations of sastrugi in the field. I agree that in general it is tempting to conclude that there must be lower accumulation where you see sastrugi in the field (I have seen this myself on lower accumulation sides of ice rises), so I think it is reasonable to mention this earlier in the manuscript, but I don't think it belongs here as one of your main conclusions.

> *We agree with the reviewer and removed this sentence from the conclusion.*

**Technical Comments**

Title: To my reading the 'the' before 'Blåskimen Island' is unnecessary. Indeed, in the abstract the name of the ice rise appears without the 'the'. *Page 1*

> *Corrected.*

L7: Suggest replacing the sentence starting with "Radar stratigraphy…" with "Arches in radar stratigraphy observed with radar suggest that the summit of the ice rise has been stable for ~600 km." This avoids introducing the concept of characteristic time in the abstract, which may be confusing to those not familiar with it.

> *Corrected.*

L11: Is there a reference for the statement about 74% of the coastline being surrounded by ice shelves?

*It is mentioned in Bindschadler et al. (2011). We cite this reference in the revised manuscript:*

*Bindschadler, R., Choi, H., Wichlacz, A., Bingham, R., Bohlander, J., Brunt, K., Corr, H., Drews, R., Fricker, H., Hall, M., Hindmarsh, R., Kohler, J., Padman, L., Rack, W., Rotschky, G., Urbini, S., Vornberger, P. and Young, N.: Getting around Antarctica: new high-resolution mappings of the grounded and freely-floating boundaries of the Antarctic ice sheet created for the International Polar Year, The Cryosphere, 5(3), 569–588, doi:10.5194/tc-5-569-2011, 2011.*

*L14: 'are' → 'is'.*

*Corrected.*

L15: Suggest delete 'eventual'.

*The sentence was removed in the revised introduction.*

L16: 'the ice shelf' → 'ice shelves'.

*Corrected.*

L16-17: this sentence about buttressing seems to repeat what was said in the previous paragraph.

*Agreed. We make this point only once in the revised introduction. (P1L19) in the marked manuscript.*

L23: Two more recent papers that use isochrones to constrain millennial scale ice-rise evolution that could be referenced here are:

Drews, R., Matsuoka, K., Martín, C., Callens, D., Bergeot, N. and Pattyn, F., 2015. Evolution of Derwael ice rise in Dronning Maud Land, Antarctica, over the last millennia. *Journal of Geophysical Research: Earth Surface*, *120*(3), pp.564-579.

and

Kingslake, J., Martín, C., Arthern, R.J., Corr, H.F. and King, E.C., 2016. Ice-flow reorganization in West Antarctica 2.5 kyr ago dated using radar-derived englacial flow velocities. *Geophysical Research Letters*, *43*(17), pp.9103-9112.

*Good point. We are aware of these references, but didn't include them in the first manuscript. Now these references are cited.*

*Page 2*

L1: Suggest replace 'using an ice core from the summit' with 'using ice cores drilled through ice rises'.

*Corrected.*

L4: insert comma after '(Fig. 1a)'.

*Corrected.*

L12: Is it narrower by a factor of two and slower by a factor of two or does the factor of two only apply to the 'slower'? Suggest switch order to '…slower by a factor of two and narrower…' if appropriate, to avoid confusion.

*We meant that ice flows slower by a factor of two and narrower. We corrected as suggested.*

L16: Reword the last sentence. 'Whereas towards east…' doesn't read well to me.

*We revised the last two sentences in this paragraph. The revised ones are (P2L32):*

*"Thus towards east, Blåskimen Island is surrounded by much slowly moving ice, whereas it abuts the eastern shear margin of the fast flowing part of the Jelbart Ice Shelf."*

L19: 'summer' → 'summers'.

*Corrected.*

*Page 3*

L3: 'were' → 'was'

*Not changed. As stakes are plural.*

L4: How many were lost and why were they lost? Were they buried? Also did you have a criterion for rejecting stakes which were tilted too much? Surely all stakes are tilted slightly. If so, it's not correct to say here that you rejected stakes that were tilted.

*We lost seven stakes out of 97 stakes. Six of them were in the southeastern slope. Stakes nearby these lost stakes show that this area has highest SMB in our study area. Stakes were 2 m high above the snow surface when they were installed and they were nearly buried one year after. So we speculate that these stakes were likely buried. One tilted stake was found in the northwestern slope, where snow surface is rough due to sastrugi. They were tilted more than ~20°, whereas the rest of stakes were tilted much smaller (only degrees or less). We then rejected this tilted stake from further analysis.*

*We have rephrased the sentence to be clearer; the revised sentence is (P3L18):*

*"The position of each of the stakes was measured in January 2013. In January 2014 we remeasured the positions of the 90 of these stakes. Six stakes were buried and one found heavily tilted (more than ~20°)."*

L5: Suggest move 'relative' to between 'stakes' and 'to'.

*Corrected.*

L6: Suggest 'infer approximate' → 'estimate'.

*Corrected.*

L13: Suggest'to the sea level' → 'to heights above local sea level'.

*Corrected.*

L13: This reads as if you used the gravity product to subtract the value. Suggest reword to say that you subtracted the value that was supplied by the gravity product.

*The new sentence is (P3L28):*

*"To convert heights above the WGS84 ellipsoid to the sea level, we subtracted 13 m of geoid height uniformly provided by GOCE gravity product (https://earth.esa.int/web/guest/dataaccess/browse-data-products/-/article/goce-gravity-fields-5777)."*

L19: 'profiles' → 'surveys'. It looks like you did more than two profiles.

*Corrected.*

L21: Delete 'a'.

*Corrected.*

L22: Delete commas after 'radar'.

*Corrected.*

L23: Insert 'a' before dewow and Ormsby. Also 'Post processing was made…' does not read well.

*Corrected. The revised sentence is (P4L9):*

*"Post processing included using a dewow filter, an Ormsby band-pass filter, and depth-variable gain functions."*

L24: 'the' →'a' and insert 'of' after 'speed'.

*Corrected.*

L25: How is the firn correction computed? Using the firn cores presumably, but this is not mentioned here.

*We added the following to text to explain this point (P4L11):*

*"We used an optimization inversion routine (Brown and Matsuoka, in prep) to model depth profiles of density along shallow radar profiles using (1) surface densities measured at 9 locations, and (2) depth profile of the density measured along the 23-m-long firn core. We then used the depth profiles of the density along the radar profile to estimate laterally variable firn corrections. The magnitudes of the firn correction vary between 4–6 m."*

L29: Transported the samples where?

*We transported the frozen samples to laboratories for further analysis. We rephrased the sentence as (P4L18):*

*"We drilled a 23 m lomg firn core near the summit of Blåskimen Island and transported the frozen samples to laboratory to develop visual stratigraphy as we well as chemical, isotope and dielectric stratigraphy."*

L30: Suggest delete 'backed'.

*Replaced 'backed' with 'back'.*

*Page 4*

L1: 'to' → 'in', insert 'the' between 'from' and 'surface'.

*Corrected.*

L5: delete 'of'.

*Corrected.*

L9-10: suggest delete 'giving a relative dome-shaped topography to the ice rise'.

*We did not delete this sentence, as we wanted to emphasize the overall shape of the ice rise.*

L11: Not clear what 'a flatter basin northeast' means and it is grammatically incorrect. I cannot see a clear distinction between slopes in the northeast and in the other directions. Also I can't see clear distinctions between basins in figure 2a.

*To clarify these issues, we revised the sentences as (P11L7):*

*"The eastern flank shows locally steep slopes and a basin in northeast with overall lower, less-tilted surface. The eastern steep slopes and the southwest ridge are consistent with lineations observed in satellite imagery (light gray feature over dark gray in Fig. 1b)."*

L12: The line 2-2' isn't strictly a flow line as it passes through the ice divide. I guess it is two flow lines connected together, but describing it as a flow line I don't think is correct.

*Replaced flowline to profile.*

L17: The estimate of the vertical uncertainty of ±5m could be more fully explained here. The center frequency of the deep radar was 2 MHz, corresponding (I think) to a wavelength of $c/n_i/2 \times 10^6 = 84.2$ m, where c is the speed of light and $n_i$ is the refractive index of ice. This wavelength is considerably more than your estimated uncertainty in digitizing the bed reflector. Is the higher precision achieved due to the signal being quite broadband? Perhaps this can be explained in more detail, as one might expect a bed reflector imaged with an 84 m wavelength radar to manifest as a layer thicker than 5 m.

*The center frequency of the deep-sounding radar is 2 MHz, giving the wavelength in ice approximately 84 m. Therefore, this radar is not capable to distinguish two objects that are separated less than 42 m (half of the wavelength). Nevertheless, this radar is capable to detect the range to the target more precisely. We sampled returned wave at 100 MHz, or every 10 ns. Over this period, radio wave travels about 2 m.*

L27: Suggest delete 'below'.

*Corrected.*

L28: The surface velocity measurements are described as a surface velocity field here, when they really appear as just point measurements in figure 2c, rather than a field.

*We used large number of point measurements of the ice-flow vector to estimate the flow field. We think that the manuscript is clear enough.*

L28: Suggest 'from' → 'of'.

*Corrected.*

*Page 5*

L9: Is this surface density or the density averaged over the top 3 m of all the cores? It is confusing because the previous sentence mentions "surface density variations", but this sentence only says 'firn density'.

*We now define the density averaged over the top 3 m (measured using 3-m-long firn cores) surface density at P4L28 and consistently use it throughout the manuscript to avoid confusion.*

L10: Suggest replace 'To estimate..' with 'When estimating SMB below…', or similar.

*Corrected.*

L17: It is stated here that *most* SMB values lie between the first and third quartiles, but doesn't this by definition mean that half the measurements lie between these two values (i.e. half ≠ most)?

*The reviewer is correct. The sentences are improved*

*from*

*SMB varies by a factor of 3.3 over the study area of ~20 km by ~20 km (0.28–1.03 m a⁻¹), although most SMB values calculated with this method range between 0.73 and 0.87 m a⁻¹ (first and third quartiles, respectively).*

*to*

*SMB varies by a factor of 3.3 over the study area of ~20 km by ~20 km (0.28–1.03 m a⁻¹), with 80% of values ranging between 0.69 and 1.03 m a⁻¹.*

L22: I am not sure I see the relevance here of the parentheses about wind direction.

*Previous studies show that SMB is largely related to the local slope (e.g. King (2004)), as wind is faster over a steeper slope. We didn't see this relationship clearly in our data (Fig. 2a) but we think that this is because the wind direction is oblique to the profile we sampled. With the text in parenthesis we want to clarify that as the slope in discussion is not aligned along the prevailing wind direction. Therefore, the observed result of 'no clear relationship' does not necessarily disagree with previous studies. No change was made in the manuscript.*

*Page 6*

L6: Is it not more precise to say that it is the larger vertical strain rates near the surface at the divide (rather than strong horizontal variations in vertical strain rates) due to the Raymond effect that could mean that the shallow-layer approximation may not be valid at the divide. Gillet-Chaulet et al. (2013) and

Kingslake et al. (2014) and probably others have measured higher magnitude vertical strain rates near the surface on ice divides and could be cited here to support the point.

Gillet Chaulet, F., Hindmarsh, R.C., Corr, H.F., King, E.C. and Jenkins, A., 2011. In situ quantification of ice rheology and direct measurement of the Raymond Effect at Summit, Greenland using a phase sensitive radar. *Geophysical Research Letters*, *38*(24).

Kingslake, J., Hindmarsh, R.C., Aðalgeirsdóttir, G., Conway, H., Corr, H.F., Gillet - Chaulet, F., Martin, C., King, E.C., Mulvaney, R. and Pritchard, H.D., 2014. Full depth englacial vertical ice sheet velocities measured using phase sensitive radar. Journal of Geophysical Research: Earth Surface, 119(12), pp.2604-2618.

*We agree with the reviewer. We included the suggested references and changed the text accordingly. The revised text are from P6L29-P6L34:*

*"However, this assumption may not be valid in areas where vertical strain rates are large, such as the region near an ice-flow divide (Gillet et al., 2011; Kingslake et al., 2014). In this region, accumulated effects of variable vertical strain can result in upward arches in isochrones, so-called Raymond arches (Raymond, 1983), which were found at many other ice rises (e.g. Vaughan et al., 1999; Conway et al., 1999). Such upward arches can also be caused by anomalously low SMB near the summit possibly due to wind erosion (Drews et al., 2013, 2015).*

Eqn 2: This equation appears incorrect. The *z* on the top of the fraction in the integrand on the right side shouldn't be there.

*We apologize for the typo in the equation. The correct equation is*

$$t(z) = \int\limits_{0}^{z} \frac{2}{v(z)} dz$$

L30: I do not see why using a constant density (when in fact the density varies with space) would make the problem ill-posed. It may make the MB inaccurate, but why ill-posed?

With this sentence, we wanted to say that the constant density assumption will make the MB calculation inaccurate. The sentence has been corrected as (P12L23):

*"It implies that the use of a uniform density could make the SMB estimates less accurate, and variable density should be accounted for."*

L32: Is there a word missing from near the start of this sentence?

*We added "for" so that the sentence starts with*

*"To account for density variations."*

*Page 7*

L19-20: There is a comparison made here between SMB derived with two different methods. In my specific comments above I suggest that the second of these methods be removed from the manuscript, but here I would suggest that if both methods must be included that the comparison is made clearer here. As figure 4 stands, we have to compare spatial patterns in the two fields, but much of the structure in these

fields appears to depends on the details of the interpolation, i.e. how the surface is interpolated across the large gaps between data points. The statement here that the two methods 'give nearly identical spatial patterns in SMB' doesn't appear to be correct, but the differences that stand out in figure 4 are due to the interpolations. So, I suggest that you include a plot that compares the two datasets without including an interpolated field. For example, a profile along the line 2-2' of the SMB estimates, or remove the spatial dimension entirely and include a scatter plot the estimates from each method on each axis.

> *While making the comparison between the two SMB estimates derived from radar stratigraphy we only used the data along the radar profiles and did not use interpolation data. Hence the comparison is free of any interpolation effects. After considering your comment above, we have decided to use the first radar SMB estimate for the mass balance calculation. Although we still mention the second radar method for discussion purposes. Based on your suggestion we also included a comparison of the two methods in Fig 3a, as well as the stake SMB measurements along the profile 2 – 2'.*

L30-31: There is a missing link here. Is the point that if there were basal melting then the Raymond arches would be smaller? As the arches heights have not been compared to arch heights expected from modelled (Martin et al., 2006) or measured (Kingslake et al., 2016) vertical velocities, I am not sure that their size can be used to support the statement about melting. Note that I think it is fine to assume that the ice rise is cold based if the 1D modelling shows that it is thin enough.

Kingslake, J., Martín, C., Arthern, R.J., Corr, H.F. and King, E.C., 2016. Ice flow reorganization in West Antarctica 2.5 kyr ago dated using radar derived englacial flow velocities. *Geophysical Research Letters*, *43*(17), pp.9103-9112.

> *Yes, this was the point. The presence of arches infer that there has not been any large amount of ice melt below the divide. However,, as you pointed out, past small scale melting cannot be ruled out without further analysis. So, as per your suggestion we have removed the statement.*

*Page 8*

The term laminar flow is used at least three times here to distinguish the flow of the ice rise flanks from flow within a region close to the divide where the Raymond effect acts. All glacial ice flow is laminar (as opposed to turbulent), so this seems doesn't seem like the correct term to use. Perhaps a better way to make the distinction is to say that in the flanks the shallow-ice approximation is valid whereas in the divide region it is not. The approximation of γ in line 7 comes from the SIA and Martin et al. (2009) showed that the full-stokes models are required to describe the Raymond effect and discuss how the SIA is incapable of this. So perhaps this works better.

Martin, C., Hindmarsh, R.C. and Navarro, F.J., 2009. On the effects of divide migration, along ridge flow, and basal sliding on isochrones near an ice divide. Journal of Geophysical Research: Earth Surface, 114(F2).

> *We agree and have made relevant changes to the paragraphs explaining the range of γ, and cite Martin et al. (2009): For changes please refer the paragraphs (P8L31 to P9L5) of the marked manuscript.*

L26-27: Can you expand on the statement that a more accurate determination of γ requires us to know ice flow history? Is this because ice rheology has a memory through ice temperature and crystal fabric?

*Yes. We have rephrased the sentence to (P9L19):*

*"More accurate determination of γ requires the knowledge of ice-rise evolution in the past millennia, because ice rheology has a memory through ice temperature and crystal fabric, and the use of detailed ice-flow modelling, which is beyond the scope of this study."*

*Page 9*

L1-2: This sentence, starting 'This is because… ' is very unclear.

*We rephrased the sentences:*

*We divide the ice rise into 19 polygons in respect to the slope direction (Fig. 6a). This is because the SMB has a contrasting feature between southeast and northwest sides (Fig. 4), and the corresponding difference in the mass balance can happen.*

*With (P9L30):*

*"As ice rises are expected to show slope dependent SMB features (King (2004), Lenaerts et al., 2014), it is probable that mass balance could also have similar features. To account for this, in this method we divide the ice rise into 19 polygons in respect to the surface slope direction (Fig. 6a) and data availability."*

I found section 6.2.3 very difficult to understand. For example, in line 21, is 'this estimate' the estimate of SMB using the flowband setup or the estimate of the variation in flowband width. Also is the second paragraph (which is just one sentence) missing more material? I suggest that the description of the flowband setup is re-written and expanded to make this clearer.

*We rephrased the paragraph (P10L17-P10L28) as:*

*"In this method, we calculate mass balance along ice-flow bands of varying widths in three different slopes of the ice rise. We define a flowband width to account for flow divergence and convergence along the flowband, assuming that ice flows along the steepest descent path on the surface. Flowband width at the downslope end is taken as 1 km, and for each flowband, steepest ascent paths are determined from two points 0.5 km away from the most downstream GPS stake. We used ascent because the surface topography near the summit is much less distinct and consequently the divergence estimate is more sensitive to small topographic changes. We rejected three flow profiles out of six, because the GPS markers were not within the defined flowband. Along the three flowbands shown in Fig 6c, the flowband widths vary by a factor of 1.4–3.6 along each flowband. This variation depends on the initial band width (1 km used here), but over the range of the initial band width between 0.9 and 2.5 km, the band width estimates vary only ~3%. We further divided the flowbands into three columns based on the available data and calculated mass balance similarly to the polygon setup.."*

L6: I am confused by this statement that for a given γ the polygon setup gives the largest estimate of mass balance. According to Fig 7, when γ > 0.75 the flowband setup (dashed line) is higher than the polygon setup (solid line).

*Here we are comparing the mass balance values averaged over all columns in a setup for a given γ. To clarify this point, the sentence is revised to (P13L15):*

*"We averaged mean mass balance values of all ice columns for each setup and for each γ (shown with point symbols in Fig. 7). For a given γ, the polygon setup give the largest estimate, whereas the grid-setup estimate is smaller by 0.02- 0.03 m a⁻¹."*

L7: Suggest remove 'much'. I am not sure that I would describe 0.05-0.07 m/a as *much* smaller than 0.1 in this context. They are the same order of magnitude.

*You are right. We removed "much".*

L8-9: I am not sure that I understand the sentence starting 'Higher γ….'. Are you saying that the dependence of the mass balance on γ is nearly linear. If so, I suggest you replace 'uniform' with 'linear'.

*The sensitivity of mass balance to γ is represented by the slope of the curves in Figure 7, which are essentially uniform for all columns . The sentence is left unchanged. (P13L20).*

L11: Delete 'as it goes'.

*Corrected.*

L12: Rephrase this sentence to avoid the phrase 'varies in a more variable way'.

*Sentence rephrased as (P13L23):*

*"Along slopes C, E and F, mass balance increases monotonically as it goes downstream, whereas mass balance of polygons along A, B and D slopes is more variable."*

*Page 11*

The discussion contains many typos and ungrammatical sentences, that I have not listed in detail.

*We apologize that the manuscript has many typos. We carefully read the manuscript to remove typos and grammatical errors in these sections.*

L15-26: Do these two paragraphs belong in the introduction? They do not discuss any of the new results and I think they can be shortened without losing substance.

*We think that these paragraphs better fit to Discussion, rather than Introduction. This is because this knowledge (mechanisms to determine SMB) is not required as background information to understand the contents of this paper. Rather, this knowledge is more useful to discuss and interpret our results in a larger context. We have shortened them considerably as (P14L28 – P15L12).*

*Page 12*

L20: Another recent relevant reference is:

Kingslake, J., Martín, C., Arthern, R.J., Corr, H.F. and King, E.C., 2016. Ice flow reorganization in West Antarctica 2.5 kyr ago dated using radar derived englacial flow velocities. *Geophysical Research Letters*, *43*(17), pp.9103-9112.

*Thanks for drawing our attention to this work. We have added this reference at P15L18.*

L26: 'inferring' → 'implying'.

*Corrected.*

L27: '(1000s m)' is confusing as throughout the manuscript spaces between values and units have not been consistent. Suggest replace with '1000's of meters'.

*Corrected.*

L29: Delete 'the' before 'Raymond'.

*Corrected.*

L31: Is a word missing from near the beginning of this sentence?

*The original sentences*

*If the summit position is steady for longer time, then the Raymond arches are further developed into double-peaked arches (Martín et al., 2009), which are not clearly observed here.*

*are changed to (P16L15):*

*If the summit position stays stable for longer time, then the Raymond arches are further developed into double-peaked arches (Martín et al., 2009), which are not clearly observed here.*

*Page 13*

Section 7.5 contains no discussion of your results, only the location of the ice rise. Can any of your findings (for example bed topography) contribute to this discussion of the impact of the ice rise on the adjacent ice shelves? If not, perhaps this belongs in the introduction or the section in which you introduce the ice rise.

*We want to discuss the role of this ice rise in current setting for completion. Discussion seems to be the right place, as the reader is now well informed of the ice rise. To make it fit better we point out how the questions raised in this paragraph will be answered in future modelling study of Blåskimen Island. The new paragraph with the added sentence (underlined) is (P16L20-P16L29):*

*Roles of ice rises vary largely in terms of its settings, and thus can change during the evolution (Matsuoka et al., 2015). Blåskimen Island is currently situated at the calving front of the local ice shelves and in the ice-flow shadow of Novyy Island ice rise upstream (Fig 1). This setting implies that currently Blåskimen Island alone has limited impact on the continental grounding line and ice flux from the ice sheet. However, it seems likely that Blåskimen Island plays a more significant role than the Novyy Island to maintain the current calving front position. Favier and Pattyn (2015) demonstrated that an ice shelf landward of the ice rise is thicker than the seaward ice shelf facilitating formation of rifts and ice-shelf breakups just seaward of the ice rise. To explore the dynamics of this surrounding region better we will use the datasets presented in this study to model the ice evolution of the ice rise.*

L17: Insert 'a' between 'over' and 'flat'.

*Corrected.*

*Figures and captions*

Figure 4 caption: The second sentence appears to be missing something.

*Replaced.*

*Circles show the locations of 90 stakes.*

*With*

*Circles show the locations of the installed stakes.*

Figure 5 caption: The end of second sentence is ungrammatical.

*We replaced the sentence:*

*Dots show the arch amplitudes and dashed lines show their linear fits to the depth.*

*With*

*"Dots show the arch amplitudes and dashed lines show the linear fits of the arch amplitudes to depth."*

Figure 7 caption, lines 3: Replace 'as' with 'to'.

*Corrected.*

**Responses to reviewer #2**

**General Comments**

This manuscript provides exhaustive details on the field measurements and data analysis over an ice rise in Dronning Maud Land, Antarctica from radar and GPS. While the manuscript provides sufficient data to support their conclusions it falls short of providing a compelling story. I think the main problem may be more due to writing however, than mis-interpretation of the data. Indeed, it seems to be more of a field report than a scientific paper that tells a story. I would prefer for the authors to follow the more traditional style of writing with sections called: introduction, data and methods, results, discussion, conclusions. While these sections are in the text, there are numerous other sections and subsections with detailed headings making it difficult to follow the logical order of the manuscript. Further, the writing could be tightened by combining very short paragraphs and reducing redundant information. For example, instead of presenting info on the ice core records in section 3, simply refer to the reference in your discussion of your measured SMB. Same for temperature info - why is this relevant to the current study? This kind of thing led to my inability to understand what the story is here.

*Thank you for your constructive comments. To emphasize more on the story we have reworked the introduction section of the manuscript. We have also reorganized the manuscript to follow a more conventional structure as suggested. Please see the new structure presented at the beginning of our response letter.*

*We keep descriptions of the 23-m-firn core in the revised manuscript because this core is used to develop depth profiles of density, which is used to derive SMB. Amongst all field measurements we made, only firn temperature is not discussed in this paper. Rather than leaving a single measurement out from this paper, we prefer to include all field measurements together in this paper so that this paper can be a comprehensive reference point when we write following papers including the one currently under development to decipher evolution of this ice rise using numerical modeling.*

**Technical Comments**

Page 3, Line 21: delete "a"

*Corrected.*

Page 3, Line 30: If you're using the firn cores to get estimates of density variability it might be nice to include a figure of these data.

*Core locations are shown in Fig. 4a (wheel spoke markers).*

Page 5, Line 8: This sentence is not needed as it was already said in a previous section.

*Agreed. We have removed this sentence.*

Page 6, Line 6-7: Don't forget the work of Nereson and Raymond, 2001

*Thank you for this suggestion. We have included this reference (P6L32).*

Page 7, Line 7: delete "a"

*Deleted.*

Page 7, Line 25: Rignot and Kanagaratnam is a better reference since they (I believe) pioneered the IO method.

*Thank you for clarifying. We have included the reference you suggested (P8L18).*

Page 7, Line 28: Not sure you can assume no melting based on the study by Neumann et al., (2008) which is on the other side of Antarctica.

*We cite Neumann et al. (2008) as a reference of the modeling method used here. This is why Neumann et al. (2008) is mentioned immediately after "one-dimensional thermomechanical model", rather than at the end of the sentence.*

Page 11, Lines 3-9: The authors make an offhand assertion that current DEMs of Antarctica do not properly resolve this ice rise elevation, nor the details contained within their DEM. I think they would have to demonstrate that the added detail is necessary in order to "get the precip right" according to

Lenaerts (2014). I might be convinced that gross differences between DEMs are important but I'm not convinced that we need to include every small detail as obtained by a ground survey.

*We regret that this paragraph can be read in this way. We intended to say that topography of ice rises is much steeper than the other parts of Antarctica and thus in general ice rises are difficult places to obtain precise elevations using satellite altimetry techniques, though they are most useful to generate continent-wide DEM. With the best of our knowledge, there is no validation study to examine continent-wide DEMs in the coastal Antarctica, focusing to ice rises. It is quite interesting to compare climate model results for different topography to test required accuracy of topography to precisely model SMB, density, etc over the ice rises and ice shelves. However, such modeling work is beyond the scope of our paper. Below, we show a revised paragraph with which we believe that our point is clearer.*

*"Distinct topographic features of the ice surface revealed with our DEM are not fully represented in continent-wide DEMs (e.g. Bamber et al., 2009; Fretwell et al., 2013, in which spatial resolution is 1 km). Also, the summit heights in those products are 24–40 m lower than our measurements. It remains unclear how much this inaccurate description of topography affects modeling SMB and surface density. Lenaerts et al. (2014) demonstrated that elevated topography associated with ice rises causes orographic precipitations and corresponding precipitation shadow not only over ice rises but also on adjacent ice shelves. Such variations could result in anomalous firn density over the ice shelves, which would result in ill-posed estimate of freeboard thickness and its long-term changes."*

Page 11, Line 31-32: your reported slope differences are within the expected range of the model prediction by Lenaerts et al, (2014) not "smaller than"

*Corrected. Rephrased the sentence as (P15L13):*

*On Blåskimen Island, we found that upwind slopes have 2–3 times the SMB on the downwind slopes, which is within the model prediction (2–4 times; Lenaerts et al. (2014)).*

Page 12, Line 19: I don't see the need for these references here - we already know what Raymond arches are.

*You are right. The reference has been removed.*

Page 12, Line 21: refer to Nereson's work on this

*Added Nereson et al. (1998) at (P16L5).*

Page 12, Line 30-32: I can clearly see double-peaked arches in Fig. 3c.

*We argue that the second arch on the side is made by a small bed undulation there. Our preliminary results of ice-flow modeling show that this arch is reproduced even if ice is assumed isotropic, which will be reported in another paper that we are developing on this manuscript. To clarify this point, we revised the sentences to (P16L15):*

*"If the summit position is stable for longer time, then Raymond arches are further developed into double-peaked arches (Martín et al., 2009), which are not clearly observed here (small side arch is caused by bed bump nearby, according to our initial ice-flow modeling)."*

Below we list the citations not shown properly in the marked manuscript due to an issue with Latexdiff –

[revised manuscript text omitted]

radar reflectors profiled with the shallow-sounding radar. Both methods require  surface-density distribution measurements.

~~We collected 3-m-long cores at 9 locations to measure firn density variations (locations are shown in Fig. 4a). Measured firn density varies by ± ~2.5% over the ice rise, with a mean value of 453 kg m$^{-3}$ (uncertainty: 3% or 14 kg m$^{-3}$). However, no distinct pattern in firn density variation was observed in terms of elevation or slope direction. While estimating SMB below, we bilinearly interpolated the firn density.~~

[revised manuscript text omitted]

~~Figure 4b shows the SMB averaged over the past 9 years (2005–2014) estimated using the second radar method. It gives the spatially mean value of 0.75 m a$^{-1}$,692 kg m$^{-2}$ a$^{-1}$ with the first and third quartiles of 0.67 and 0.85 m a$^{-1}$ 611 and 784 kg m$^{-2}$ a$^{-1}$ respectively. The first radar method gives a slightly higher number than the second radar method by 5–10%; the mean value for the first method is 0.81 m a$^{-1}$, 744 kg m$^{-2}$ a$^{-1}$ with first and third quartiles of 0.71 and 0.93 m a$^{-1}$. 650 and 860 kg m$^{-2}$ a$^{-1}$ The difference between the SMB estimated with these two methods is localized to the region in 1-6 km northeast of the summit. Except for this northeast region, these two methods give nearly identical spatial patterns in SMB.~~

[revised manuscript text omitted]

~~Ice rises in DML receive most of their precipitation during sporadic synoptic events occurring 0–5 times per year (?). In such events, moisture-rich air from the Southern Ocean approaches the ice rises from a northeasterly direction. The orographic lifting of air on the ice rise leads to precipitation on the northeast slope and a precipitation shadow on the southwesterly slope. During no-precipitation periods, this region is dominated with southeasterly katabatic wind (Lenaerts et al., 2014). Consequently, the observed SMB distribution is a result of precipitation events with the northeasterly wind and erosion/redistribution with the southeasterly wind.~~

[revised manuscript text omitted]

---

## Referee Report (RR1)

Review of: Glaciological settings and recent mass balance of Blåskimen Island in Dronning Maud Land, Antarctica, Goel and others, tc-2017-61

This paper provides a study of the glacial conditions, largely concerning mass balance, of one of Antarctica's larger ice rises, which is situated in the DML sector. Most of the content is a straightforward report of a field measurement campaign, where much of the motivation is to assess the ice rise's mass balance. The main conclusions are that the mass balance of the ice rise has been positive over the last decade, and that the ice rise has likely been stable for at least the last ~600 years.

The paper certainly presents data that are worth publishing and that will interest a number of readers. Down the line the data should provide a useful resource for calibrating regional mass balance measurements. The paper also represents a further case study of ice-rise conditions to complement the recent work on Derwael and Halvfarryggen. For all of these reasons it is a useful contribution.

A previous version of the paper was reviewed by two different reviewers, and the response has certainly moved the paper towards a form that is better structured and more compelling. My comments are, therefore, largely directed towards improving the efficiency and correctness of the writing.

**Comments**

***Universal changes throughout manuscript:***

There is no need for the word "the" in front of e.g. Fimbul Ice Shelf, Jelbart Ice Shelf. Additionally, when used as proper nouns, e.g. in Fimbul Ice Shelf, Jelbart Ice Shelf, or even Fimbul and Jelbart Ice Shelves, then Ice Shelf/Shelves should be capitalised. Use lowercase when referring to ice shelf/shelves more generally.

You introduce the acronym SMB for surface mass balance multiple times, and even sometimes use the full words again rather than taking advantage of having introduced the acronym early in the paper. Please go back through the manuscript sticking to this convention:
  - Use surface mass balance (full words) in abstract.
  - Introduce acronym SMB at first opportunity in main text (as you have done on P1 L16)
  - Be consistent thereafter in using only the acronym SMB
  - Optionally, you might choose to keep the full words "surface mass balance" in Figure captions and Section 7 Conclusions, catering for readers who might skim-read in the first instance.

Please avoid split infinitives. Currently there are examples at P2 L3, P3 L10. You can search for these comprehensively by searching for the word "to".

Anywhere where you describe e.g. ice sheet versus ice-sheet thinning, ice shelf versus ice shelf thinning, ice rise versus ice-rise elevation, and so on, you should not use a hyphen when just referring to e.g. the ice sheet, but you should introduce the hyphen into it when describing e.g. the ice-sheet thinning. Most of the time you follow this rule well, but there are exceptions, and I recommend you search these out, and correct/confirm on a case-by-case basis. You might do this by using the search function for "ice" (and "radar", as, for example, in radar-detected features).

You interchange the terms ice divide, flow divide and ice-flow divide. You could clean this up a bit by using a single term.

Some of the writing in Section 4 oscillates between being written in the past tense and the present tense. It would generally be better written consistently in the past tense.

***Figure ordering***

I suggest you reverse the order of Figures 2 and 3. My reasoning is explained in the line by line comments.

***Line by Line***

Title. I am not sure "glaciological setting" is really capturing what you do, nor is it that compelling as part of a paper's title. Would the paper be better titled: "Mass balance and stability of Blåskimen Island, Dronning Maud Land, Antarctica"? Much of the paper's focus is essentially on the present and past mass balance, with stability in the title representing the elements of the study that investigate whether the mass balance has changed significantly over time.

I am not convinced the abstract really sells the paper all that well, nor captures the full logic of the paper. Suggested restructure:
First sentence: East Antarctica's Dronning Maud Land ice-shelf-fringed coast contains numerous ice rises that influence the dynamics and mass balance of the region.

P1, L12: Insert commas: "…shelves, together…"   "…rumples), regulate…"

P1, L17-18: Start with: "Hence, although…"   insert "areal" before "footprint" and remove *the* in L18.

P1, L20: Rewrite as "…evolution of ice rises and adjacent ice bodies over…"

P1, L23-24: More efficiently phrased as: "…Kingslake et al., 2016), supplementing ice-core-derived climate records (e.g. Mulvaney et al., 2002, 2014).

P2, L1: Capital L: Land

P2, L3: Better to substitute shape with dynamics here.

P2, L6: plural, beds

P2, L7: plural, contrasts, emplace comma after slopes

P2, L8: "…that both of these ice rises…"

P2, L8-9: "…5000 years), despite being separated by…"   "and ranging across variable glaciological settings."

P2, L12: This final sentence of the paragraph would be better written along the lines of: "These existing observations underscore the requirement for further detailed investigations of ice rises in DML."

Section 2. I felt this section could be written with a more finessed logic. I suggest the very first sentence is: "Blåskimen Island (Fig. 1a; total area 651 km$^2$; Moholdt and Matsuoka, 2015) is the most seaward of a series of isle-type ice rises (totally surrounded by floating ice) and promontory-type ice rises (elongated extensions of the ice sheet into an ice shelf) that partitions Jelbart Ice Shelf from Fimbul Ice Shelf.
Then discuss Jelbart Ice Shelf, fed by Schytt Glacier, then Fimbul Ice Shelf, fed by Jutulstraumen but buttressed near the western calving front near Blåskimen by the further ice rises/rumples.
Then finish with a sentence along the lines of: "In summary, ice flow to the south and north of Blåskimen Island is slow; ice flow to its east is also slowed by the ice rises and rumples on the western shear margin of the otherwise fast-flowing Fimbul Ice Shelf; and hence the fastest flow near Blåskimen Island occurs to its west where it abuts the eastern shear margin of Jelbart Ice Shelf.

P3, L2-4: Conflate to a single sentence, requiring one instance of replacing surveys with measurements: "We carried out field measurements of…" "…2013-2014, comprising kinematic…" "…(Section 3.2), and firn coring…"

P3, Opening section. I think this small opening paragraph would benefit from a small expansion explaining WHY you gathered each of the named datasets. Currently readers are only being fed this information later in the manuscript.

P3, L10: Introduce comma: "…ice rise, and…"

P3 P15: More efficiently phrased as: "…(Matsuoka et al., 2012b), and 90 reoccupied in January 2014, the remaining 7 being lost to snow burial or found to be tilting by > 20°. Note that this no longer specifies that 6 were buried and one tilted but I suspect readers have negligible interest in that specificity.

P3 L21: Remove comma: "…kinematic and…"

P3 L22: …using TRACK software, part of the GAMIT/GLOBK GPS package (Herring…"

P3 L29: sp. descent

P3 L31: Insert "a": "…operated a GSSI…"

P3 L32: "…Hawley et al., 2014). Both radar surveys…"

P4 L13: I think "record" is more suitable here than "develop."

P4 L21: "Hereafter we refer to mean density as…"

P4 L24: "…estimating SMB below 3 m depth, we…

Section 4.1
Nowhere as the 2x methods for determining SMB are introduced is there a helpful and immediate clarification that the two methods have relative merits in terms of the simplicity of the measurements versus the timescales of information they address. I'd like it to be made more upfront in the opening to this section that the stake method is undertaken because it can inform quickly on the broad patterns of change between 2013 and 2014, albeit with an overall uncertainty of 6%, but that the radar method can give a longer-term view.

P4 L31: "…accounting for snow…"

P5 L5: "…reflectors is solely…"

P5 L6: "Thus the shallow-ice…" "…when the depth…"

P5 L11: "…which have been found…"

P5 L12: Substitute "Such upward" with "Raymond"

P5 L14: "…demonstrated that the amplitude of Raymond arches…"

P5 L16: There's no need to refer to Figure 2 here. The main effect is to distract the reader. It seems that your purpose in referring to Figure 2 at this point is simply to illustrate a typical Raymond arch, but I strongly suspect your target readership knows what this is already. If they don't, they should be referring to Raymond 1983!

P5 L26 and L28: Replace "is" with "was".

P5 L28: "method is to invert simultaneously for spatial variations in…" and cut all words from altogether to the citation.

The only explanation (as such) you give for the inversion is to refer to an in prep manuscript. There is no description of the technique given at all. This issue was also raised by Reviewer#1 first time around, and I don't think you have addressed it. If you are going to use results from this technique at all in the paper, there needs to be some more information on it. Is the Brown/Matsuoka inversion at least part derived from an earlier method that can be cited here?

P6 L8: "…melting, as a one-dimensional…"

P6 L9: The author's full surname(s) – admittedly an unusual form – is Fox Maule (n.b. no hyphen). Thus it should appear as Fox Maule et al here, and should be listed in the reference list as Fox Maule, C.

P6 L9: I think it would be appropriate to insert a word like "Moreover," to start this sentence.

P6 L10: I think that here, as with P5 L16, there's not a big need to refer to Fig. 2 yet. The flow of the paper would be improved by having people first look at the results figures in the results section.

P6 L13: Insert and: "…column, and γ.."

P6 L18: "…over a non-sliding bed and using the shallow-ice…"

P6 L20-21: "…Drews et al., 2015), giving γ between…"

P6 L17-L28: Make all this a single paragraph.

P6 L22-23: Suggest: "However, because the ice is not isothermal and, near the ice divide, the shallow-ice approximation is also invalid, in reality the range of γ is wider."

P6 L31: "For the thermomechanical…"

P7 Section 4.2.2. I would consider this section to be more logically ordered throughout as (1) flowband method, (2) polygon setup, and (3) grid setup. The first represents the most direct reliance on the field measurements, the other two rely more on interpolation to expand the coverage. I note that in your results (Section 5) you follow this order in the text, so it would be better streamlined writing to discuss the method in the same order here.

P8 L14: Here, if you agree with my earlier suggestions that there has not yet been a need to refer readers to the radargrams in Fig. 2, then it would be better to order the current figures so that Fig. 2 is now the 3 panel figure of surface/bed/velocities. Hence change text here to refer to Fig. 2a.

P8 L20 and Fig. 3b, now suggested to be Fig. 2b: Perhaps add the profile numbers 1-1', 2-2' etc as also marked on Fig. 1b. Figs. 2 and 3 will be located close to each other in the final paper, so this might save readers having to flick back to Fig. 1 to locate the radargram positions. If you do this, then you should change the locational reference in the text to "2-2' in Figs. 1b and 2b".

P8 L21, L22: Here it now makes logical sense that you introduce readers for the first time to the plots of surface slope and radargrams respectively. But they would now be labelled Fig. 3a, b, c.

Below this point in the paper, swap all references to Figs 2 and 3.

P9 L13: Not really sure why it's especially specified here that the study area is 20 x 20 km. You've already stated the area in Section 2, and a reader can see the area on Figs. 1b, 3, 4. It just seems to make the sentence overlong. There does, however, seem to be a discrepancy between the 20 x 20 km of these figures and the value of 651 km$^2$ written in P2 L19 (and in P11 L9). How is the smaller area of 651 km$^2$ defined specifically?

P9 L14: Is it not more correct to note that the main contrast is between the northwest and the southeast?

P9 L23: "…are controlled primarily by SMB versus the Raymond effect…" "we measured the amplitudes…"
P9 L24: Extra "the" needed x2: "We used the two deeper arches of the three that we used for SMB…"

P9 L29: represent (not represents)

P10 L16-19 and Figure 6. I suggest rearranging Fig. 6 panels so that panel a is flowband, panel b polygon and panel c grid. This would make all of Section 4.2, 5.3 and Fig. 6 consistently ordered. You could easily introduce some more specific references to Fig 6a, b and c where relevant in L16-19.

P10 L21-23: This reads like a sentence that should have been used back in Section 4.2.

P10 L25: gives

P10 L29: southeasternmost?

P10 L31-32: I'm not sure I follow what you mean by thickening and thinning in this sentence. Does this refer to the trend for mass balance to increase/decrease upslope/downslope in slopes A and F? I see that in slope A the lowermost polygon has a much lower mass balance than the upper two polygons in slope A, but I do not see the reverse trend in slope F – in slope F the mass balance looks pretty similar in each polygon. I see that the average mass balance of slope A is lower than the average mass balance of slope F, although whether the average mass balance is actually the lowest of all slopes is not clear, because slope D has low values too. Similarly, it's not clear (if this is what you're trying to say) that slope F has the highest mass balance of all slopes, at least from the information one can draw from Figure 6.

P11 L1: Suggest rephrasing: "Together, the measurements show that Blåskimen Island had positive mass balance between 2005 and 2014."

P11 L2-4: Another opportunity to re-order.

P11 L4: Use the symbol for gamma as you have done elsewhere.

P11 L13-L24: There are some missing words in the current sentence but in any case there's no real need to specify that your DEM comes from GPS at this stage of the paper. Suggest: "Our detailed surface DEM (Fig. 2a) reveals a number of surface topographic features that are smoothed over in continent-wide DEMs (e.g. Bamber et al., 2009; Fretwell et al., 2013). It confirms, for example, that the lineations in satellite imagery observed in satellite imagery over Blåskimen Island correspond to surface undulations (c.f., Goodwin and Vaughan, 1995)" N.b., the c.f. is important, because Goodwin and Vaughan didn't refer to this location. Start the final paragraph essentially with a rewritten version of the current paragraph's second sentence: "We further note that the summit height of Blåskimen Island is 24-40 m higher in our DEM compared with the lower resolution DEMs"
…. However, can you clarify for sure that this is not a consequence of the different products being referenced to different vertical datums?
I think it is disingenuous to describe the Bamber/Bedmap2 as an "inaccurate description of topography". It's lower resolution, which is not the same thing at all.

P11 L28: This is an inappropriate use of "inferring". You could replace it with: "from which we infer"

P11 L29: vary should be varies.

P12 L1: Suggest: "…SMB has been observed on other Antarctic ice rises. For example, King (2004) showed…"

P12 L7: "The net impact…" "…coast has been examined using the RACMO2…"

P12 L16: "…Derwael Ice Rise, where it was attributed to wind…"

P12 L29: Here, if you do not already have data or a different publication to cite that gives the modelling results, you would be better advised to write: "…which are likely Raymond arches, though ice-flow modelling would be required to confirm this interpretation."

P13 L13-L20: I'm not convinced this paragraph is really saying anything that couldn't have been said in the absence of all your new data. Certainly the final sentence is inappropriate for a published paper.

P13 L22: Past tense: "investigated"

Figure 1
Ice Shelf should be written with capital letters in both labels on Fig. 1a.
Since you mention Schytt Glacier in the main text this may also be worth marking on Fig. 1a.

Figure 4
Did you think about producing a difference map as a third panel, to help in the general comparison of the results?

---

## Author Response (AR2)

**Response Letter**

**Dear Editor,**

We thank the reviewers for their constructive comments. We have considered all suggestions carefully. Please see our responses below. We regret that the previous versions included language issues; the revised manuscript that we are submitting now was proof-read by an English editing service. Overall, we believe that this version meets journal's criteria but we are also willing to revise the manuscript if it is not the case.

In this letter, we refer page and line numbers in the marked manuscript.

Thanks for handling our manuscript.

Vikram Goel, on behalf of coauthors

**Review #1**

Review of: Glaciological settings and recent mass balance of Blåskimen Island in Dronning Maud Land, Antarctica, Goel and others, tc-2017-61

This paper provides a study of the glacial conditions, largely concerning mass balance, of one of Antarctica's larger ice rises, which is situated in the DML sector. Most of the content is a straightforward report of a field measurement campaign, where much of the motivation is to assess the ice rise's mass balance. The main conclusions are that the mass balance of the ice rise has been positive over the last decade, and that the ice rise has likely been stable for at least the last ~600 years.

The paper certainly presents data that are worth publishing and that will interest a number of readers. Down the line the data should provide a useful resource for calibrating regional mass balance measurements. The paper also represents a further case study of ice-rise conditions to complement the recent work on Derwael and Halvfarryggen. For all of these reasons it is a useful contribution.

A previous version of the paper was reviewed by two different reviewers, and the response has certainly moved the paper towards a form that is better structured and more compelling. My comments are, therefore, largely directed towards improving the efficiency and correctness of the writing.

We thank the reviewer for investing time on this manuscript. We found all the comments very constructive and helpful. To make sure we do not leave any more grammatical issues we used an English editing service for this revision.

**Comments**

**Universal changes throughout manuscript:**

There is no need for the word "the" in front of e.g. Fimbul Ice Shelf, Jelbart Ice Shelf. Additionally, when used as proper nouns, e.g. in Fimbul Ice Shelf, Jelbart Ice Shelf, or even Fimbul and Jelbart Ice Shelves,

then Ice Shelf/Shelves should be capitalised. Use lowercase when referring to ice shelf/shelves more generally.

**Understood and corrected for. Please refer to marked manuscript for the corrections.**

You introduce the acronym SMB for surface mass balance multiple times, and even sometimes use the full words again rather than taking advantage of having introduced the acronym early in the paper. Please go back through the manuscript sticking to this convention:

- Use surface mass balance (full words) in abstract.
- Introduce acronym SMB at first opportunity in main text (as you have done on P1 L16)
- Be consistent thereafter in using only the acronym SMB
- Optionally, you might choose to keep the full words "surface mass balance" in Figure captions and Section 7 Conclusions, catering for readers who might skim-read in the first instance.

The manuscript has been updated as per your suggestion. In the revised manuscript, we define SMB at P1L17 in Introduction. However, we use "surface mass balance" instead of SMB in the Abstract, section titles, and conclusions.

Please avoid split infinitives. Currently there are examples at P2 L3, P3 L10. You can search for these comprehensively by searching for the word "to".

**We followed this suggestion.**

Anywhere where you describe e.g. ice sheet versus ice-sheet thinning, ice shelf versus ice shelf thinning, ice rise versus ice-rise elevation, and so on, you should not use a hyphen when just referring to e.g. the ice sheet, but you should introduce the hyphen into it when describing e.g. the ice-sheet thinning. Most of the time you follow this rule well, but there are exceptions, and I recommend you search these out, and correct/confirm on a case-by-case basis. You might do this by using the search function for "ice" (and "radar", as, for example, in radar-detected features).

We followed this suggestion and made the changes. Further, the submitted manuscript has been reviewed by an English editing service.

You interchange the terms ice divide, flow divide and ice-flow divide. You could clean this up a bit by using a single term.

*Corrected. We refer to it as 'ice-flow divide' once and as 'divide' afterwards in the manuscript. P5L31 onwards*

Some of the writing in Section 4 oscillates between being written in the past tense and the present tense. It would generally be better written consistently in the past tense.

We have checked Section 4 (Analytical methods) to ensure that all descriptions of our past activities are in the past tense. Statements about things such as definitions and known facts are in the present. In some cases, we were not consistent originally, and these cases have been corrected.

**Figure ordering**

I suggest you reverse the order of Figures 2 and 3. My reasoning is explained in the line by line comments.

After carefully considering your suggestion, we have decided to keep the figure order the same. This is because the clear detection of the bed shown in Figure 2 provides base knowledge to generate Fig. 3 (bed DEM).

**Line by Line**

Title. I am not sure "glaciological setting" is really capturing what you do, nor is it that compelling as part of a paper's title. Would the paper be better titled: "Mass balance and stability of Blåskimen Island, Dronning Maud Land, Antarctica"? Much of the paper's focus is essentially on the present and past mass balance, with stability in the title representing the elements of the study that investigate whether the mass balance has changed significantly over time.

We agree with the reviewers' comment that there is scope for improvement in the manuscript's title. We agree with the first half of the reviewer's suggestion, but not the second half. We discuss stability of the ice rise in the manuscript, but it is only one of the main conclusions that we have made. We present ice thickness, topography, and SMB, which are all new knowledge from this region. Stability is within the scope of this paper but without numerical modeling of ice flow it is hard to rigorously conclude the stability of the ice rise. That is why we put the stability issues in discussion, not results. Our primary focus is the mass balance of the ice rise, among other glaciological characteristics. We tried and consider other alternative titles such as -

Mass balance and other characteristics of Blåskimen Island Ice Rise DML, Characteristics of Blåskimen Island Ice Rise DML, Recent thickening of Blåskimen Island Ice Rise DML etc.

But, at the end of this exercise we couldn't think of a better alternative than the present title.

I am not convinced the abstract really sells the paper all that well, nor captures the full logic of the paper. Suggested restructure:

First sentence: East Antarctica's Dronning Maud Land ice-shelf-fringed coast contains numerous ice rises that influence the dynamics and mass balance of the region.

We have largely followed the suggestion about the first sentence. But after some review, we decided not to make other major changes to the abstract.

P1, L12: Insert commas: "...shelves, together..." "...rumples), regulate..."

**Corrected. P1L13**

P1, L17-18: Start with: "Hence, although..." insert "areal" before "footprint" and remove the in L18.

**Corrected. P1L18**

P1, L20: Rewrite as "...evolution of ice rises and adjacent ice bodies over..."

Corrected. P1L22

P1, L23-24: More efficiently phrased as: "...Kingslake et al., 2016), supplementing ice-core-derived climate records (e.g. Mulvaney et al., 2002, 2014).

Corrected. P1L24

**P2, L1: Capital L: Land**

Corrected. P2L3

P2, L3: Better to substitute shape with dynamics here.

Corrected. P2L5

P2, L6: plural, beds

Corrected. P2L9

P2, L7: plural, contrasts, emplace comma after slopes

Corrected. P2L9

P2, L8: "...that both of these ice rises..."

**Corrected. P2L10**

P2, L8-9: "...5000 years), despite being separated by..." "and ranging across variable glaciological settings."

**Rephrased as (P2L10) -**

"Stratigraphic evidence shows that both of these ice rises have been in nearly steady state over the last several millennia (3000–5000 years), despite being separated by ~1200 km along the coast with variable glaciological settings."

P2, L12: This final sentence of the paragraph would be better written along the lines of: "These existing observations underscore the requirement for further detailed investigations of ice rises in DML."

**Corrected. P2L16**

Section 2. I felt this section could be written with a more finessed logic. I suggest the very first sentence is:
 "Blåskimen Island (Fig. 1a; total area 651 km2; Moholdt and Matsuoka, 2015) is the most seaward of a series of isle-type ice rises (totally surrounded by floating ice) and promontory-type ice rises (elongated extensions of the ice sheet into an ice shelf) that partitions Jelbart Ice Shelf from Fimbul Ice Shelf.
 Then discuss Jelbart Ice Shelf, fed by Schytt Glacier, then Fimbul Ice Shelf, fed by Jutulstraumen but

buttressed near the western calving front near Blåskimen by the further ice rises/rumples.

Then finish with a sentence along the lines of: "In summary, ice flow to the south and north of Blåskimen Island is slow; ice flow to its east is also slowed by the ice rises and rumples on the western shear margin of the otherwise fast-flowing Fimbul Ice Shelf; and hence the fastest flow near Blåskimen Island occurs to its west where it abuts the eastern shear margin of Jelbart Ice Shelf.

The section was rewritten largely following the suggestions above. (P2L24 – P3L9).

"Blåskimen Island (Fig. 1; total area 651 km2 (Moholdt and Matsuoka, 2015)) is the most seaward of a series of isle-type ice rises (surrounded by floating ice or ocean) that partitions Jelbart Ice Shelf from Fimbul Ice Shelf. Jelbart Ice Shelf is fed by the Schytt Glacier, which is slower by a factor of two and narrower than the Jutulstraumen Glacier that feeds into Fimbul Ice Shelf (Rignot et al., 2011). Jutulstraumen Glacier, one of the largest outlet glaciers in DML (Høydal, 1996) is buttressed towards its western calving front by four small ice rises and rumples near Blåskimen Island. In summary, ice flow to the south of Blåskimen Island is slow, with open ocean to the north; ice flow to its east is also slowed by the ice rises and rumples on the western shear margin of the otherwise fast-flowing Fimbul Ice Shelf. As a result, the fastest flow near Blåskimen Island occurs to its west where it abuts the eastern shear margin of Jelbart Ice Shelf. This setting implies that currently Blåskimen Island alone has limited impact on the continental grounding line and ice flux from the ice sheet. However, as Favier and Pattyn (2015) demonstrated how an ice rise aids the formation of rifts and ice-shelf breakups on its seaward side, Blåskimen Island likely plays a more significant role than upstream ice rises to maintain the current calving-front position."

P3, L2-4: Conflate to a single sentence, requiring one instance of replacing surveys with measurements: "We carried out field measurements of..." "...2013-2014, comprising kinematic..." "...(Section 3.2), and firn coring..."

This paragraph has rephrased as (P3L11) -

"To estimate the mass balance of the ice rise, we made field measurements on Blåskimen Island during the austral summers of 2012–2013 and 2013–2014. The measurements included kinematic and static GPS surveys (Section 3.1), shallow- and deep-sounding radar profiling (Section 3.2), as well as firn coring and borehole temperature measurements (Section 3.3). The location of these measurements is shown in Fig. 1b."

P3, Opening section. I think this small opening paragraph would benefit from a small expansion explaining WHY you gathered each of the named datasets. Currently readers are only being fed this information later in the manuscript.

An opening line has been added to the paragraph and some other minor changes were made (P3L11).

"To estimate the mass balance of the ice rise, we made field measurements on Blåskimen Island during the austral summers of 2012–2013 and 2013–2014. The measurements included..."

P3, L10: Introduce comma: "...ice rise, and..."

The sentence has been rephrased. P3L21 -

"To locate the ice rise's summit, we ran additional surveys near the summit and in the eastern part of the ice rise where satellite imagery shows surface lineations (light grey feature over dark grey in Fig. 1b)."

P3 P15: More efficiently phrased as: "...(Matsuoka et al., 2012b), and 90 reoccupied in January 2014, the remaining 7 being lost to snow burial or found to be tilting by > 20°. Note that this no longer specifies that 6 were buried and one tilted but I suspect readers have negligible interest in that specificity.

Rephrased as suggested (P3L23) -

"The stakes were occupied for ~20 minutes to determine their lateral positions (e.g., Conway and Rasmussen, 2009; Matsuoka et al., 2012b), and 90 were reoccupied in January 2014, the remaining 7 being lost to snow burial or found to be tilting by > 20°."

P3 L21: Remove comma: "...kinematic and ... "

Corrected. P4L4

P3 L22: ...using TRACK software, part of the GAMIT/GLOBK GPS package (Herring..."

Corrected. P4L5

P3 L29: sp. descent

Corrected. P4L13

P3 L31: Insert "a": "...operated a GSSI ... "

Corrected. P4L15

P3 L32: "...Hawley et al., 2014). Both radar surveys..."

**Corrected. P4L16**

P4 L13: I think "record" is more suitable here than "develop."

**Corrected. P4L29**

P4 L21: "Hereafter we refer to mean density as..."

Corrected. P5L7

P4 L24: "...estimating SMB below 3 m depth, we...

Sorry that you misunderstood. We meant to refer to the SMB calculation later in the manuscript. The sentence has been corrected to be clearer. P5L11 –

"We then used the bilinearly interpolated surface density to estimate SMB."

**Section 4.1**

Nowhere as the 2x methods for determining SMB are introduced is there a helpful and immediate clarification that the two methods have relative merits in terms of the simplicity of the measurements versus the timescales of information they address. I'd like it to be made more upfront in the opening to this section that the stake method is undertaken because it can inform quickly on the broad patterns of change between 2013 and 2014, albeit with an overall uncertainty of 6%, but that the radar method can give a longer-term view.

Added the following sentences to as per your suggestion (P5L16) -

"The stake method is simpler and provides insight on SMB pattern over the ice rise for the period of 2013-2014. The radar method has higher uncertainties, but provides a longer-term view of past SMB patterns."

P4 L31: "...accounting for snow..."

Corrected. P5L20

P5 L5: "...reflectors is solely..."

Corrected. P5L26

P5 L6: "Thus the shallow-ice..." "...when the depth..."

The sentence has been rephrased. P5L27

"With these assumptions, the shallow-layer approximation holds when the depth h (ice equivalent) of a radar reflector is much smaller than the local ice thickness H (h<<H)."

P5 L11: "...which have been found ... "

Corrected. P5L33

P5 L12: Substitute "Such upward" with "Raymond"

Replaced 'such' with 'similar' as the observed shallow arches are similar to Raymond arches but are not the same. P6L1

P5 L14: "...demonstrated that the amplitude of Raymond arches..."

**Corrected. P6L3**

P5 L16: There's no need to refer to Figure 2 here. The main effect is to distract the reader. It seems that your purpose in referring to Figure 2 at this point is simply to illustrate a typical Raymond arch, but I strongly suspect your target readership knows what this is already. If they don't, they should be referring to Raymond 1983!

The observed upward arches in the shallow radar are not the same as Raymond arches due to a different mechanism of formation, although they are both 'upward arches'.

In these lines (P5L31 – P6L5) we are discussing and comparing ('similar arches') these distinct features observed in shallow and deep radar profiles. We feel while reading these lines, the reader will be curious to see these features first hand and that is why we cite figure 2 here.

For completeness, we cited Fig 2c (P5L32) when we first talk about Raymond arches.

P5 L26 and L28: Replace "is" with "was".

Sentence rephrased as (P6L18) -

"The first method accounts only for vertical variations in density and ignores any lateral variations."

P5 L28: "method is to invert simultaneously for spatial variations in..." and cut all words from altogether to the citation.

Sentence rephrased as (P6L19) -

"The second method involves simultaneously inverting for spatial variations in density, temperature, and SMB (Brown and Matsuoka, in prep)."

The only explanation (as such) you give for the inversion is to refer to an in prep manuscript. There is no description of the technique given at all. This issue was also raised by Reviewer#1 first time around, and I don't think you have addressed it. If you are going to use results from this technique at all in the paper, there needs to be some more information on it. Is the Brown/Matsuoka inversion at least part derived from an earlier method that can be cited here?

We have tried to clarify this method in more detail through these sentences – (P6L19) – There is no prior work using an optimization inversion route in our way so we didn't add a reference.

"The second method involves simultaneously inverting for spatial variations in density, temperature, and SMB (Brown and Matsuoka, in prep). It uses an optimization inversion routine to solve for the best fit between a steady state firn density model (Herron and Langway, 1980) and the measured two-way travel times to multiple isochrones identified in shallow-radar profiles. Our optimization routine is constrained by surface densities measured at 13 locations along the shallow-radar profile as well as the measured depth- density profile along the 23 m long firn core."

P6 L8: "...melting, as a one-dimensional..."

Sentence rephrased as (P7L5) -

"As a one-dimensional thermomechanical model (Neumann et al., 2008) shows no basal melt for the geothermal-flux estimate in this region (57 mW m-2; Fox Maule et al., 2005), we ignored basal melting."

P6 L9: The author's full surname(s) – admittedly an unusual form – is Fox Maule (n.b. no hyphen). Thus it should appear as Fox Maule et al here, and should be listed in the reference list as Fox Maule, C.

*Corrected, both text and the citation. P7L6 (not highlighted as LatexDiff has trouble with citation changes)*

P6 L9: I think it would be appropriate to insert a word like "Moreover," to start this sentence.

Added. P7L7

P6 L10: I think that here, as with P5 L16, there's not a big need to refer to Fig. 2 yet. The flow of the paper would be improved by having people first look at the results figures in the results section.

We agree that it is not necessary to mention figure 2 here in the method section. But, as we have already referred to this figure once by this point in the manuscript, we prefer to continue to refer to it here for the benefit to the curious reader.

The figure 2 was not switched with figure 3 and was cited early in the method section because of two reasons -

- As the clear detection of the bed shown in Figure 2 is the base knowledge to generate Fig. 3 (bed DEM).
- As Figure 2 is cited at a point where we are discussing two kinds of upward arches which look similar but formed differently. And reader could benefit looking at figure 2 at that point.

P6 L13: Insert and: "...column, and γ.."

Corrected. P7L11

P6 L18: "...over a non-sliding bed and using the shallow-ice..."

Corrected. P7L17

P6 L20-21: "...Drews et al., 2015), giving γ between..."

Corrected. P7L20

P6 L17-L28: Make all this a single paragraph.

Corrected. (P7L16 – P7L29) (It is not showing properly in the marked manuscript due to a bug in the Tex software, but unmarked version has these as one paragraph.)

P6 L22-23: Suggest: "However, because the ice is not isothermal and, near the ice divide, the shallow-ice approximation is also invalid, in reality the range of y is wider."

We mention these points, though write it a little differently (P7L21-P7L21) -

"However, due to ice-temperature variations and ice-divide effects, the latter of which invalidate the shallow-ice approximation, the range of  $\gamma$  should be wider."

P6 L31: "For the thermomechanical..."

Corrected. P7L32 – P8L3

"Also within 10H of the divide, the case of thermomechanical flow gives  $0.69 \le \gamma \le 0.86$  Hvidberg (1996) and the case of isothermal axisymmetric radial flow gives  $0.54 \le \gamma \le 0.76$  Hvidberg (1996)."

P7 Section 4.2.2. I would consider this section to be more logically ordered throughout as (1) flowband method, (2) polygon setup, and (3) grid setup. The first represents the most direct reliance on the field measurements, the other two rely more on interpolation to expand the coverage. I note that in your results (Section 5) you follow this order in the text, so it would be better streamlined writing to discuss the method in the same order here.

We agree and reordered the section as suggested. Now we mention flowband setup first, polygon setup second, and grid setup last (P8L21 – P8L19)

P8 L14: Here, if you agree with my earlier suggestions that there has not yet been a need to refer readers to the radargrams in Fig. 2, then it would be better to order the current figures so that Fig. 2 is now the 3 panel figure of surface/bed/velocities. Hence change text here to refer to Fig. 2a.

We agree and changed the order of figure panels accordingly; (a) flowband, (b) polygon, and (c) grid setups.

P8 L20 and Fig. 3b, now suggested to be Fig. 2b: Perhaps add the profile numbers 1-1', 2-2' etc as also marked on Fig. 1b. Figs. 2 and 3 will be located close to each other in the final paper, so this might save readers having to flick back to Fig. 1 to locate the radargram positions. If you do this, then you should change the locational reference in the text to "2-2' in Figs. 1b and 2b".

We have marked the radar profiles in the Figure 3b as suggested.

P8 L21, L22: Here it now makes logical sense that you introduce readers for the first time to the plots of surface slope and radargrams respectively. But they would now be labelled Fig. 3a, b, c.

As we responded above, we decided not to change the figure order, because the clear detection of the bed shown in Figure 2 is the base knowledge to generate Fig. 3 (bed DEM). As well as Figure 2 is cited at a point where we are discussing two kinds of upward arches which look similar but formed differently. And reader could benefit looking at figure 2 at that point.

Below this point in the paper, swap all references to Figs 2 and 3.

P9 L13: Not really sure why it's especially specified here that the study area is 20 x 20 km. You've already stated the area in Section 2, and a reader can see the area on Figs. 1b, 3, 4. It just seems to make the sentence overlong. There does, however, seem to be a discrepancy between the 20 x 20 km of these figures and the value of 651 km2 written in P2 L19 (and in P11 L9). How is the smaller area of 651 km2 defined specifically?

The area of the ice rise is indeed  $651 \text{ km}^2$ . The term  $20 \times 20 \text{ km}$  was used to give a first order idea during talks etc and should not have been included in the manuscript. Now removed. (P11L5)

P9 L14: Is it not more correct to note that the main contrast is between the northwest and the southeast?

Yes. Now updated. (P11L7)

P9 L23: "...are controlled primarily by SMB versus the Raymond effect..." "we measured the amplitudes..."

Corrected. Now rephrased as (P11L16) -

"To judge whether the reflector depths are controlled primarily by SMB or the Raymond effect..."

P9 L24: Extra "the" needed x2: "We used the two deeper arches of the three that we used for SMB..."

Corrected in the rephrased sentence (P11L17) -

"Due to the shallowest reflector having insignificant amplitude, we used just the two deeper reflectors of the three that were used for SMB estimates."

P9 L29: represent (not represents)

Yes. P11L24

P10 L16-19 and Figure 6. I suggest rearranging Fig. 6 panels so that panel a is flowband, panel b polygon and panel c grid. This would make all of Section 4.2, 5.3 and Fig. 6 consistently ordered. You could easily introduce some more specific references to Fig 6a, b and c where relevant in L16-19.

Figure 6 panels haven been rearranged.

P10 L21-23: This reads like a sentence that should have been used back in Section 4.2.

Agree. Now moved to Section 4.2 (P8L18).

**P10 L25: gives**

**Corrected. P12L26**

**P10 L29: southeasternmost?**

**Yes. Corrected. P12L33**

P10 L31-32: I'm not sure I follow what you mean by thickening and thinning in this sentence. Does this refer to the trend for mass balance to increase/decrease upslope/downslope in slopes A and F? I see that in slope A the lowermost polygon has a much lower mass balance than the upper two polygons in slope A, but I do not see the reverse trend in slope F – in slope F the mass balance looks pretty similar in each polygon. I see that the average mass balance of slope A is lower than the average mass balance of slope F, although whether the average mass balance is actually the lowest of all slopes is not clear, because slope D has low values too. Similarly, it's not clear (if this is what you're trying to say) that slope F has the highest mass balance of all slopes, at least from the information one can draw from Figure 6.

We apologize for the confusion. After updating figure 6 with the different SMB dataset after the first review, the slope F does not show the high SMB in the lower slopes as it did before. We should have updated the text accordingly after the update in the dataset. Now this sentence has been removed. The updated paragraph (P12L31 – P13L5)-

"For polygon and grid setups, all the columns show positive mass balance over the full γ range, except for southeasternmost downstream polygon A3 (the slope-direction codes are shown in Fig. 6b). Along slopes C, E and F, mass balance does not vary significantly along the slope, whereas mass balance of polygons along slopes A, B, and D is more variable. For the Flowband setup, six of nine columns show positive mass balance, with columns CD3 and DE1 (see Fig. 6a for the slope-direction codes) being very close to balance. Column DE3 has negative mass balance in the northwest downstream, a region where the estimated flow divergence is anomalously large."

P11 L1: Suggest rephrasing: "Together, the measurements show that Blåskimen Island had positive mass balance between 2005 and 2014."

We rephrased the sentence as (P13L6) -

"In summary, the measurements show that Blåskimen Island had positive mass balance between 2005 and 2014."

P11 L2-4: Another opportunity to re-order.

**Done. P13L7-P13L11**

P11 L4: Use the symbol for gamma as you have done elsewhere.

Yes, Corrected. P13L10

P11 L13-L24: There are some missing words in the current sentence but in any case there's no real need to specify that your DEM comes from GPS at this stage of the paper. Suggest: "Our detailed surface DEM (Fig. 2a) reveals a number of surface topographic features that are smoothed over in continent-wide DEMs (e.g. Bamber et al., 2009; Fretwell et al., 2013). It confirms, for example, that the lineations in satellite imagery observed in satellite imagery over Blåskimen Island correspond to surface undulations (c.f., Goodwin and Vaughan, 1995)" N.b., the c.f. is important, because Goodwin and Vaughan didn't refer to this location. Start the final paragraph essentially with a rewritten version of the current paragraph's second sentence: "We further note that the summit height of Blåskimen Island is 24-40 m higher in our DEM compared with the lower resolution DEMs"

.... However, can you clarify for sure that this is not a consequence of the different products being referenced to different vertical datums?

I think it is disingenuous to describe the Bamber/Bedmap2 as an "inaccurate description of topography". It's lower resolution, which is not the same thing at all.

Thank you for pointing out the possible issue; we re-visited the issue more carefully. First, different geoid models are used for previous work, which should be corrected accordingly. However, different geoid datasets cannot be the main reason of the apparent difference; geoid heights of the cell nearest to Blåskimen Island are 14.4 m (GL04C used in BEDMAP2/Bamber DEM), and 13 m (GOCE, used in our work). Second, different continent-wide DEMs describe the shape and height of Blåskimen Island differently. For example, BEDMAP2 surface elevations match well at the summit with the data (difference: 10 m), but they are quite different in the flank (100 m). Bamber DEM is offset from the observations by 80-100 m. These numbers do not reflect statistical significance of these DEMs and we do not assess uncertainties of these DEMs using these numbers. However, we want to briefly mention such possibly large difference over ice rises and emphasize the need of field data to better calibrate or validate DEMs. To reflect these points, the text was updated as follows (P13L20-P14L6)-

"Our detailed surface DEM (Fig. 3a) reveals a number of surface topographic features that are smoothed over in continent wide DEMs. For example, two widely-used DEMs (Bamber et al., 2009; Fretwell et al., 2013) show different topography of the ice rise and elevations are off from our local DEM by 10-100 m at different places. Ice rises have much steeper slopes than the continental slope (Fig. 2a), which inherently requires high-spatial resolutions to accurately represent the topography. This missing detail could affect modelling SMB and surface density. Lenaerts et al. (2014) demonstrated that elevated topography associated with an ice rise causes orographic precipitation and the resulting precipitation shadow not only over the ice rise, but also on the adjacent ice shelves. Such variations could result in anomalous firn-density estimates over the ice shelves, which would result in ill-posed estimates of freeboard thickness and the resulting long-term changes of adjacent ice shelves. Our local DEM confirmed that lineations in satellite imagery over Blåskimen Island appear where the surface slope has greater variability, with most lineations being associated with uneven bed topography. Such an association was originally proposed over Fletcher Promontory Ice Rise by (Goodwin and Vaughan, 1995). This agreement supports the use of satellite imagery as a remote means to explore first-order surface and bed topography."

P11 L28: This is an inappropriate use of "inferring". You could replace it with: "from which we infer"

Corrected. P14L10

P11 L29: vary should be varies.

Sentence rephrased (P14L11) -

"SMB averaged over this period has a mean of 0.81 m  $a^{-1}$ , but varies along the radarprofiles between 0.71 m  $a^{-1}$  (first quartile) and 0.93 m  $a^{-1}$  (third quartile)."

P12 L1: Suggest: "...SMB has been observed on other Antarctic ice rises. For example, King (2004) showed..."

Corrected. P14L14

P12 L7: "The net impact..." "...coast has been examined using the RACMO2..."

**Corrected. P14L21**

P12 L16: "...Derwael Ice Rise, where it was attributed to wind..."

**Corrected. P14L33**

P12 L29: Here, if you do not already have data or a different publication to cite that gives the modelling results, you would be better advised to write: "...which are likely Raymond arches, though ice-flow modelling would be required to confirm this interpretation."

**We agree. Now corrected. P15L15**

P13 L13-L20: I'm not convinced this paragraph is really saying anything that couldn't have been said in the absence of all your new data. Certainly the final sentence is inappropriate for a published paper.

We have removed this paragraph (P16L01- P16L8) and moved some of its content to section 2 (P3L6) where Blåskimen Island is introduced.

**P13 L22: Past tense: "investigated"**

Changed, changed the sentence to past tense as (P16L10) -

"We used geophysical methods to investigate Blåskimen Island, one of the larger isle-type ice rises at the calving front at the intersection of Fimbul and Jelbart Ice Shelves on the DML coast."

Figure 1

Ice Shelf should be written with capital letters in both labels on Fig. 1a. Since you mention Schytt Glacier in the main text this may also be worth marking on Fig. 1a.

Corrected.

Figure 4

Did you think about producing a difference map as a third panel, to help in the general comparison of the results?

The difference plot of stake-derived SMB and radar-derived SMB is shown here with Fig. 1b to present data availability from the stake and radar methods –

We feel this SMB comparison suffers with two reasons. First, the spatial coverage of these two datasets are different. For example, the maximum difference appears North, where no radar data exist. Second, both the datasets deal with different time periods.

Due to these reasons, we decided, not to include this panel in the revised manuscript.

**Review #2**

I reviewed a previous version of this manuscript. I outlined my appreciation for the importance of glaciological work on ice rises in the previous review. Here I comment on the authors' responses to my original comments (https://doi.org/10.5194/tc-2017-61-RC1). I also highlight some of the typos that are still present.

In the attached annotated copy of the response letter I have commented on the responses. Below are additional specific comments. It would be useful if the authors noted where in the revised manuscript specific revision have been made. Also, unhelpfully, there are several differences between the quoted text in the responses and the revised text. I have found many grammatical errors. But I have only documented some of those that are in sentences revised in response to my previous comments. (Page and line numbers refer to the non-marked up version of the revised manuscript).

We really appreciate the amount of constructive corrections and suggestions you provided in the first review. And we really apologize to have caused you inconvenience with our responses. We did multiple checks to our response and ended up making some last moment changes, which led to being a mismatch in the response letter and the revised manuscript. This time we have been extra

careful and have only included the quotes from the manuscript at the very end of the process. Also, we have tried to include the position of all the changes even for minor changes.

Specific comments

P317: Remove extra 'the' here.

We have rephrased the sentence as (P4L4)-

"Instantaneous kinematic and average static-rover station locations were determined relative to the base stations for each field season using TRACK software, part of GAMIT/GLOBK GPS package (Herring et al., 2010)."

P8L16-17: "...giving a relatively dome-shaped topography to the ice rise." reads poorly to me.

Corrected. (P10L6) Now " giving a dome-like shape to the ice rise "

P4L21: "Hereafter we call mean density in the top 3 m as surface density." Suggest replace 'call' with 'refer to".

Corrected as suggested. P5L7

The reviewer provided comments marked on the response letter. Below, we respond to relatively major comments first and minor ones afterwards.

Reviewer: I am not sure I see the relevance here of the parentheses about wind direction.

Response: Previous studies show that SMB is largely related to the local slope (e.g. King (2004)), as wind is faster over a steeper slope. We didn't see this relationship clearly in our data (Fig. 2a) but we think that this is because the wind direction is oblique to the profile we sampled. With the text in parenthesis we want to clarify that as the slope in discussion is not aligned along the prevailing wind direction. Therefore, the observed result of 'no clear relationship' does not necessarily disagree with previous studies. No change was made in the manuscript.

Reviewer: The point is that reader of the paper will not necessarily make this link with more detail. When the reference to wind direction comes out of no where like this, it is hard to recall the background that you describe here without some clues. More details are needed here.

This sentence is more closely related to individual mechanisms that make up the SMB. Therefore, we moved this sentence to the paragraph immediate above where we discuss wind-related mechanisms. In this way, we can separate our discussion on individual processes to make up SMB in one paragraph (P14L14-P14L22) and overall SMB distribution mostly in comparison with modeled SMB in the paragraph immediately below (P14L23-P14L33).

Reviewer: The surface velocity measurements are described as a surface velocity field here, when they really appear as just point measurements in figure 2c, rather than a field.

Response: We used large number of point measurements of the ice-flow vector to estimate the flow field. We think that the manuscript is clear enough

Reviewer: Where is this described or plotted? All I see are the point measurements. To me a field implies some kind of continuity of date coverage. Therefore I disagree that the data plotted in Fig. 3c is a flow field.

We have now replaced the word 'field' with 'velocities' and 'velocity measurements' as applicable throughout the manuscript. E.g. P2L21, P3L24, P10L2, P10L28 etc

Reviewer: The estimate of the vertical uncertainty of  $\pm 5m$  could be more fully explained here. The center frequency of the deep radar was 2 MHz, corresponding (I think) to a wavelength of c/ni/2x106 = 84.2 m, where c is the speed of light and ni is the refractive index of ice. This wavelength is considerably more than your estimated uncertainty in digitizing the bed reflector. Is the higher precision achieved due to the signal being quite broadband? Perhaps this can be explained in more detail, as one might expect a bed reflector imaged with an 84 m wavelength radar to manifest as a layer thicker than 5 m.

Response: The center frequency of the deep-sounding radar is 2 MHz, giving the wavelength in ice approximately 84 m. Therefore, this radar is not capable to distinguish two objects that are separated less than 42 m (half of the wavelength). Nevertheless, this radar is capable to detect the range to the target more precisely. We sampled returned wave at 100 MHz, or every 10 ns. Over this period, radio wave travels about 2 m.

Reviewer: This explanation makes sense, but there is no indication of where this has been explained in the revised text.

The response we provided in the letter is mainly clarifying the difference between depth accuracy and resolutions. Although it is useful information, adding this clarification in length to the manuscript distracts the flow of the paper. Therefore, we decided not to include it in the revised manuscript.

(Previous comments not included as its not relevant to the comment below)

Reviewer: In the text you now use the shallow-ice approximation (SIA), where you previously used shallowlayer approximation. SIA is a term already used to describe an approximation to the Stokes Equations for ice-sheet model which is different than neglecting vertical strain for shallow radar layers. So I support the original terminology: shallow-layer approximation.

It is our mistake to change both "laminar flow" and "shallow-layer approximation" to "shallow-ice approximation". The reviewer suggested to change only the former and it is what we wanted to make. However, in the last editing process, both of them were changed to Shallow-ice approximation. We sincerely apologize for this error. Now, shallow-ice approximation is used to explain the ice flow, but shallow-layer approximation is used to explain the assumption to derive the SMB using radar data.

**Other comments -**

This is different than in the text.

It would be useful if you said where these changes have been made in the marked manuscript

There some differences between the quoted text below and the text in the marked manuscript.

We apologize for this mismatch. We have been a lot more careful to avoid such issues this time.

replace with "towards the east" or "eastwards"

We have rephrased large part of this section. Relevant sentences at – (P2L24-P3L9)

OK. Well change the start of this sentence to "The position of each stake...." then.

delete ('the')

We have rephrased the sentence as (P3L28) -

"The stakes were occupied for ~20 minutes to determine their lateral positions (e.g., Conway and Rasmussen, 2009; Matsuoka et al., 2012b), and 90 were reoccupied in January 2014, the remaining 7 being lost to snow burial or found to be tilting by > 20°."

"We then used the depth profiles of the density along the radar profile to estimate laterally variable firn corrections."

Is a word missing here? Perhaps "with"? (between 'along' and 'the')

The sentence rephrased as – (P4L24)

"The correction was estimated using the modeled depth profiles of the density along the radar profile based on firn-core density observations and shallow-radar data, as further discussed in section 4.1."

**Delete ('we')**

**A word is missing here.**

Sentence rephrased. (P4L29) -

"To record the stratigraphy (visual, chemical, isotopic, and dielectric), we drilled a 23 m long firn core near the ice-rise summit. The core was dated back to 1996 by counting annual cycles of oxygen isotopes and by identifying volcanic horizons using non-sea-salt sulphate data (Vega et al., 2016)."

Word missing still.

Added 'the'. P10L7

replace with 'with'

Corrected. P9L4

**Glaciological settings and recent mass balance of Blåskimen Island in Dronning Maud Land, Antarctica**

Vikram Goel1, Joel Brown1,2, and Kenichi Matsuoka1 1Norwegian Polar Institute, Tromsø, Norway 2Aesir Consulting LLC, Missoula, Montana, USA *Correspondence to:* Vikram Goel (vikram.goel@outlook.com)

Abstract. The ice-shelf-fringed coast of Dronning Maud Land <del>coast</del> in East Antarctica has-contains numerous ice rises that very likely control-influence the dynamics and mass balance of this-
[revised manuscript text omitted]

---

## Author Response (AR3)

Editor Decision: Publish subject to technical corrections (09 Oct 2017) by Olaf Eisen

Comments to the Author:

Dear Virkam and coauthors,

thank you for your revisions. Given the reviewers recommendation and your response I can accept your manuscript for publication in TC. However, I have some comments still (hence techical correction), which you have to take care of before publication (page and line refer to revised version4), please see below.

Regards,

Olaf

*Dear Editor,*

*It is great to hear that the manuscript is getting close to being accepted. Thank you for suggesting further corrections. We have carefully considered all of the suggestions. Please find our response below, where we refer page and line numbers in the marked manuscript. Thanks for handling our manuscript.*

*Vikram Goel, on behalf of co-authors*

p2l15: As the readers don't have the density at hand, it would be nice to give the thickening also in terms of meters of the surface.

*We updated the thickening estimate range using surface density as (P2L15)–*

*"Our analysis of these data implies that the ice rise has thickened by 0.12–0.37 m a$^{-1}$ (ice equivalent or 0.24–0.75 m a$^{-1}$in surface elevation) over the past decade.*

p2l21: I find it puzzling that after two sentences the next one already starts with "The summary ...". I suggest to change that to "Overall, ..."

*Corrected (P2L21).*

p2l27: I find this too speculative. If you think so, than this would be your hypothesis to test - which is not done in this paper.

*We intend to make two inferences in this sentence. The first one is that Blåskimen Island has limited influence on upstream flow dynamics and the second one is that it is playing relatively important role in the calving front position. The first inference is based on the observation that Blåskimen Island is in the ice-flow shadow of Novyy Island (isle-type ice rise) upstream, limiting Blåskimen Island's influence on the upstream ice dynamics. The second inference is based on the Favier and Pattyn (2015) study which showed the presence of thinner ice shelves downstream of ice rises which are subject to breaking, whereas they provide stability to the ice shelves upstream. We feel that there is enough prior information to make such speculation. So we keep the text unchanged.*

p3l21: make sure the URL is correctly broken across linebreaks.

*Corrected (P3L20).*

p4l27: Do you have any other reason to believe this, i.e. evidence?

*No we do not have any other supporting evidence.*

p5l24: Please change the reference to a manuscript in preparation. Either to pers. comm. or something else (e.g. an abstract already presented at EGU is citeable as Geophys. Abstr. Series).

*This work is not presented elsewhere yet but we are working to prepare a manuscript. According to the journal policy, all work including "submitted to", "in preparation", "in review" or only available as preprint should also be included in the reference list. So, we followed this guidance and cited this work as "Brown and Matsuoka (in prep)". In this way, we can show responsible authors and title for this work. We think that it is better than just saying "Brown (per. Comm.)" particularly because Brown is the second author of this paper. If the house editor has a different opinion, we will accept their technical corrections.*

*https://www.the-cryosphere.net/for_authors/manuscript_preparation.html*

*Works "submitted to", "in preparation", "in review", or only available as preprint should also be included in the reference list. Please do not use bold or italic writing for in-text citations or in the reference list.*

p8l4: I suggest you use M_B for the variable to be consistent with M_{SMB}. MB does not make a good variable, as written now.

*Corrected. Replaced MB by $M_{MB}$ (P8L4).*

p9l14/15: southeast vs northeast: this cannot be right if you look at Fig. 4a. Do you mean northwest?

*Corrected (P9L15). Sorry for this mistake.*

p9l21: "at the 23 m long core site" change to "at the site of the 23 m long core".

*Corrected (P9L21).*

p9l32: vertically along its length: this is tautological. Remove vertical (as you did not drill a horizontal core).

*Corrected (P9L32).*

p13l9: T=H/SMB: you do not have a variable SMB, only M_{SMB}. Change.

*Corrected (P13L9).*

p14l28: The whole paragraph has to be adapted and the data should be available before proof reading.

*The data is now available at – https://data.npolar.no/project/19bacc06-f081-4615-ae05-12dae73345bf*

*The paragraph has been rephrased as (P14L28) –*

*The datasets used or derived in this paper are available at: https://data.npolar.no/project/19bacc06-f081-4615-ae05-12dae73345bf. It includes radar, GPS, and borehole thermistor data, as well as ice thickness, surface flow velocity, and surface mass balance datasets. The 23 m long firn-core data and their availability are described in Vega et al. (2016).*

whole manuscript: "minus" should either be in math mode (LaTex) or long hyphen (--).

*Checked and corrected throughout the manuscript.*

Fig. 1 b: I can't see a green curve, just blue.

*Corrected.*

Fig. 2: Do not start a sentence with '+', change to "The SMB derived from stake heights is indicated by `+`markers. Insert "in black" after highlighted.

*Corrected.*

Fig. 3: Insert north arrow (as you talk about cardinal directions in the text). (Polar Stereo is only has one line parallel to north.) Indicate the reference level for the heights, WGS84?

*Arrow inserted. Added - "The elevations are relative to the geoid surface."*

Fig. 4: Insert north arrow.

(a) shows much more than only the SMB from the stakes, also the interpolation.
(b) ## (a) SMB during 2013-14 interpolated from stake measurements. Circles….

*Corrected*

After (b) please rewrite:

(c) SMB estimated from radar over the past decade, accounting only for the vertical variability of density. White curves show the location of radar profiles. …

*Corrected*

Fig. 6: "are divided into 3 …" (remove blank between in and to).

add "each" after "(A-F) with 3 polygons".

*Corrected*

Non-public comments to the Author:

I can, at least in general, confirm your finding that both of the used DEM standard products have shortcomings, not only as an effect of lower resolutin. We experienced the same on another ice rise in the area.

[revised manuscript text omitted]